# Toward the Fundamental Limits of Imitation Learning

**Nived Rajaraman**
University of California, Berkeley
nived.rajaraman@berkeley.edu

**Lin F. Yang**
University of California, Los Angeles
linyang@ee.ucla.edu

**Jiantao Jiao**
University of California, Berkeley
jiantao@eecs.berkeley.edu

**Kannan Ramchandran**
University of California, Berkeley
kannanr@eecs.berkeley.edu

## Abstract

Imitation learning (IL) aims to mimic the behavior of an expert policy in a sequential decision-making problem given only demonstrations. In this paper, we focus on understanding the minimax statistical limits of IL in episodic Markov Decision Processes (MDPs). We first consider the setting where the learner is provided a dataset of $N$ expert trajectories ahead of time, and cannot interact with the MDP. Here, we show that the policy which mimics the expert whenever possible is in expectation $\lesssim \frac{|\mathcal{S}|H^2 \log(N)}{N}$ suboptimal compared to the value of the expert, even when the expert plays a stochastic policy. Here $\mathcal{S}$ is the state space and $H$ is the length of the episode. Furthermore, we establish a suboptimality lower bound of $\gtrsim |\mathcal{S}|H^2/N$ which applies even if the expert is constrained to be deterministic, or if the learner is allowed to actively query the expert at visited states while interacting with the MDP for $N$ episodes. To our knowledge, this is the first algorithm with suboptimality having no dependence on the number of actions, under no additional assumptions. We then propose a novel algorithm based on minimum-distance functionals in the setting where the transition model is given and the expert is deterministic. The algorithm is suboptimal by $\lesssim |\mathcal{S}|H^{3/2}/N$, matching our lower bound up to a $\sqrt{H}$ factor, and breaks the $\mathcal{O}(H^2)$ error compounding barrier of IL.

## 1 Introduction

Imitation learning or apprenticeship learning is the study of learning from demonstrations in a sequential decision-making framework in the absence of reward feedback. The imitation learning problem differs from the typical setting of reinforcement learning in that the learner no longer has access to reward feedback to learn a good policy. In contrast, the learner is given access to expert demonstrations, with the objective of learning a policy that performs comparably to the expert's with respect to the *unobserved reward function*. This is motivated by the fact that the desired behavior in typical reinforcement learning problems is easy to specify in words, but hard to capture accurately through manually-designed rewards [AN04]. Imitation learning has shown remarkable success in practice over the last decade - the work of [ACQN07] showed that using pilot demonstrations to learn the dynamics and infer rewards can significantly improve performance in autonomous helicopter flight. More recently, the approach of learning from demonstrations has shown to improve the state-of-the-art in numerous areas: autonomous driving [HVP+18, PCS+20], robot control [ACVB09], game AI [ILP+18, Vin19] and motion capture [MTD+17] among others.

Following the approach pioneered by [BDH+05], several works [RB10, BSH20] show that carrying out supervised learning (among other approaches) to learn a policy provides black box guarantees on the suboptimality of the learner, in effect "reducing" the IL problem to supervised learning. In particular when the expert follows a deterministic policy, [RB10] discuss the behavior cloning approach, which is a supervised learning method for imitation learning by minimizing the number of mistakes made by the learner compared to the expert under the empirical state distribution in the expert demonstrations. However, the authors conclude that supervised learning could lead to severe error compounding due to the "covariate shift problem": the actual performance of learner depends on its own state distribution, whereas training takes place with respect to the expert's state distribution. Furthermore, it remains to see how the reduction approach fares when the expert follows a general stochastic policy. As we discuss later, in this setting, the reduction analysis is in fact loose and can be improved: we instead use a novel coupling based approach to provide near optimal guarantees.

Nevertheless, the aforementioned reduction approach is quite popular in studying IL and shows that it suffices to approximately solve an intermediate problem to give one directional bounds on the suboptimality of a learner. But it is unclear whether a difficulty in solving the intermediate problem implies an inherent difficulty in solving the original imitation learning problem. In this work, we cast the imitation learning problem in the statistical decision theory framework and ask,

**What are the statistical limits of imitation learning?**

We investigate this question in a tabular, epsiodic MDP over state space $\mathcal{S}$, action space $\mathcal{A}$ and episode length $H$, one of the most basic settings to start with. The value $J(\pi)$ of a (possibly stochastic) policy $\pi$ is defined as the expected cumulative reward accrued over the duration of an episode,

$$J(\pi) = \mathbb{E}_\pi \left[ \sum_{t=1}^{H} \mathbf{r}_t(s_t, a_t) \right] \tag{1}$$

where $\mathbf{r}_t$ is the unknown reward function of the MDP at time $t$, and the expectation is computed with respect to the distribution over trajectories $\{(s_1, a_1), \cdots, (s_H, a_H)\}$ induced by rolling out the policy $\pi$. Denoting the expert's policy by $\pi^*$ and the learner's policy $\widehat{\pi}$, the *subopimality* of a learner, which is a random variable, is the difference in value of the expert's and learner's policies: $J(\pi^*) - J(\widehat{\pi})$.

In the imitation learning framework, we emphasize that the learner *does not observe rewards* while interacting with the MDP, but is given access to expert demonstrations to learn a good policy. The learner is said to "interact" with the MDP by submitting a state-action pair to an oracle and receiving the next state, with the reward being hidden. We study the problem in the following 3 settings:

(a) **No-interaction:** The learner is provided a dataset of $N$ independent rollouts of the expert policy. The learner is not otherwise allowed to interact with the MDP.

(b) **Known-transition:** The only difference compared to the no-interaction setting is that the MDP state transition functions and initial state distribution are exactly known to the learner.

(c) **Active:** The learner is not given a dataset of expert demonstrations in advance. However, the learner is allowed to actively query the expert using previous expert feedback while interacting with the MDP for $N$ episodes.

The active setting gives the learner more power than the no-interaction setting: it can just follow the observed expert actions (same as no-interaction setting), or actively compute a new policy on the fly based on expert feedback, and then use the most up-to-date policy to interact with the MDP. DAGGER [RGB11] and AGGRAVATE [RB14] are popular approaches proposed for IL in the active setting.

## 1.1 Main results

The reduction approach in [RB10] shows that if the expert policy is deterministic, and the probability of error in guessing the expert's action at each state is $\epsilon$, then $J(\pi^*) - J(\widehat{\pi}) \lesssim \min\{H, \epsilon H^2\}$. The appearance of this $H^2$ factor is called *error compounding* which can be intuitively understood as the inability to get back on track once the learner makes a mistake. Indeed the basis for this argument is that if at each time the learner makes an error at time $t$ with probability $\epsilon_t$, and gets lost thereafter, incurring an error of $H - t + 1$, the error incurred is $\lesssim H\epsilon_1 + (H-1)\epsilon_2 + \cdots + \epsilon_H \lesssim \min\{H, H^2\epsilon\}$ where $\epsilon = \frac{\epsilon_1 + \cdots + \epsilon_H}{H}$ is the probability of error averaged across time. This informal argument supports

the reduction of imitation learning to the problem of minimizing the empirical probability of error, known as "behavior cloning".

Is this error compounding inevitable or is it just a consequence of the behavior cloning algorithm? Our first contribution shows that it is *fundamental* to the imitation learning problem without additional assumptions: even if the learner operates in the active setting and the expert is deterministic, no algorithm can beat the $H^2$ barrier. The term *instance* is used to refer to the underlying MDP and the expert's policy in the imitation learning problem.

**Theorem (informal) 1.1** (Formal version: Theorem 5.1 (a))**.** *In the active setting, for any learner $\widehat{\pi}$, there exists an instance such that the suboptimality of the learner is at least $|\mathcal{S}|H^2/N$ up to universal constants, i.e., $J(\pi^*) - \mathbb{E}[J(\widehat{\pi})] \gtrsim |\mathcal{S}|H^2/N$. This lower bound applies even if the expert's policy $\pi^*$ is constrained to be deterministic.*

The key intuition behind this result is to identify that at states which were never visited during the interactions with the MDP, the learner has no prior knowledge about the expert's policy or what state transitions are induced under different actions. With no available information, the learner is essentially forced to play an arbitrary policy on these states. Indeed, for $\epsilon$ which is a function of $|\mathcal{S}|$ and $N$, we construct an instance such that for any learner, the probability of visiting at time $t$ a state unseen during the interactions is $\epsilon(1-\epsilon)^{t-1}$. Upon visiting such a state, the learner gets completely lost thereafter with constant probability, and incurs a loss of $H - t$. The suboptimality is therefore $\gtrsim (H-1)\epsilon + (H-2)\epsilon(1-\epsilon) + \cdots + \epsilon(1-\epsilon)^{H-2} \gtrsim \min\{H, H^2\epsilon\}$.

Our next result shows that if the expert is deterministic, one can in fact achieve the bound $|\mathcal{S}|H^2/N$ in the no-interaction setting by the behavior cloning algorithm:

**Theorem (informal) 1.2** (Formal version: Theorem 3.2 (a))**.** *When the expert's policy is deterministic, in the no-interaction setting, the expected suboptimality of a learner $\widehat{\pi}$ that carries out behavior cloning satisfies $J(\pi^*) - \mathbb{E}[J(\widehat{\pi})] \lesssim |\mathcal{S}|H^2/N$ on any instance.*

We prove this result exactly as stated: by first bounding the population 0-1 risk of the policy to be $|\mathcal{S}|/N$ and subsequently invoking the black box reduction [RB10, Theorem 2.1] to get the final bound on the expected suboptimality of the learner.

The optimality of behavior cloning and Theorem (informal) 1.1 point to an interesting observation: the ability to actively query the expert *does not* improve the minimax expected suboptimality beyond the no-interaction setting. An important implication of this result is that DAGGER [RGB11] and other algorithms that actively query an expert, *cannot improve over behavior cloning* in the worst case. However, we remark that our bounds are worst-case, and does not imply that behavior cloning outperforms other existing algorithms on more structured instances observed in practice.

The closest relative to our bounds on behavior cloning in the deterministic expert setting is [SVBB19] the authors of which propose the FAIL algorithm. When the expert's policy is deterministic, FAIL is suboptimal by $\lesssim \sqrt{|\mathcal{S}||\mathcal{A}|H^5/N}$ (ignoring logarithmic factors). In contrast, in Theorem 3.2 (a), we show that behavior cloning is suboptimal by $\lesssim |\mathcal{S}|H^2/N$ which always improves on the guarantee of FAIL: not only is it independent of $|\mathcal{A}|$, but has optimal dependence on $H$ and $N$. However, note that the two results apply in slightly different settings and are not directly comparable: FAIL applies in the ILFO setting where the learner does not observe the actions played by the expert in the set of demonstrations, and only observes the visited states. The ILFO setting assumes that the reward function of the MDP does not depend on the action chosen at a state.

Prompted by the success of behavior cloning in the deterministic expert setting, it is natural to ask whether supervised learning reduction continues to be a good approach when the expert is stochastic. The reduction from [RB10] indeed still guarantees that any policy with total variation (TV) distance $\epsilon$ with expert's action distribution has suboptimality $\lesssim H^2\epsilon$ (see Lemma A.4). However there is a problem with invoking such a reduction to bound the suboptimality of the learner: the empirical action distribution at each state converges very slowly to the population expert action distribution under TV distance. Seeing as it corresponds to matching the expert's and learner's policies at different states which are distributions over $\mathcal{A}$, the population risk suffers from a convergence rate dependent on the number of actions, with rate $\sqrt{|\mathcal{A}|/N}$ instead of $N^{-1}$. In order to prove tight guarantees on the expected suboptimality of a policy, we are therefore forced to circumvent the reduction framework.

We analyze the MIMIC-EMP policy in this setting, which carries out empirical risk minimization under log loss. This is a natural extension of empirical risk minimization under 0-1 loss to the

| Expert | Setting | Upper bound | Lower bound |
|---|---|---|---|
| Det. | No-interaction | $\frac{\|\mathcal{S}\|H^2}{N}$ (Thm. 3.2 (a)) | $\frac{\|\mathcal{S}\|H^2}{N}$ (Thm. 5.1 (a)) |
| | Active | $\frac{\|\mathcal{S}\|H^2}{N}$ (Thm. 3.2 (a)) | $\frac{\|\mathcal{S}\|H^2}{N}$ (Item 5.1 (a)) |
| | Known-transition | $\min\left\{\frac{\|\mathcal{S}\|H^{3/2}}{N}, \sqrt{\frac{\|\mathcal{S}\|H^2}{N}}\right\}$ (Thm. 4.1 (a)) | $\frac{\|\mathcal{S}\|H}{N}$ (Thm. 5.1 (b)) |
| Non. Det. | No-interaction | $\frac{\|\mathcal{S}\|H^2 \log(N)}{N}$ (Thm. 3.3) | $\frac{\|\mathcal{S}\|H^2}{N}$ (Thm. 5.1 (a)) |
| | Active | $\frac{\|\mathcal{S}\|H^2 \log(N)}{N}$ (Thm. 3.3) | $\frac{\|\mathcal{S}\|H^2}{N}$ (Thm. 5.1 (a)) |
| | Known-transition | $\frac{\|\mathcal{S}\|H^2 \log(N)}{N}$ (Thm. 3.3) | $\frac{\|\mathcal{S}\|H}{N}$ (Thm. 5.1 (b)) |

Table 1: Minimax expected suboptimality in different settings (bounds are up to universal constants)

stochastic expert setting. The namesake for this policy follows from the fact that minimizing the empirical risk under log loss precisely translates to the learner playing the empirical expert policy distribution at states observed in the expert dataset. It is interesting to note that when the expert is determinstic, MIMIC-EMP indeed still minimizes the empirical 0-1 risk to 0 and continues to be optimal in this setting. We show that when the expert is stochastic, the expected suboptimality of MIMIC-EMP does not depend on the number of actions. Moreover from the lower bound in Theorem (informal) 1.1, it is in fact minimax optimal up to logarithmic factors.

**Theorem (informal) 1.3** (Formal version: Theorem 3.3). *In the no-interaction setting, the expected suboptimality of a learner $\widehat{\pi}$ carrying out MIMIC-EMP is upper bounded by $J(\pi^*) - \mathbb{E}[J(\widehat{\pi})] \lesssim |\mathcal{S}|H^2 \log(N)/N$ on any instance. This result applies even when the expert plays a stochastic policy.*

The main ingredient in the proof of this result is a coupling argument which shows that the expected suboptimality of the learner results only from trajectories where the learner visits states unobserved in the expert dataset, and carefully bounding the probability of this event.

We next discuss the setting where the learner is not only provided expert demonstrations, but the state transitions functions of the MDP. The "known-transition" model appears frequently in robotics applications [ZWM$^+$18], capturing the scenario where the learner has access to accurate models / simulators representing the dynamics of the system, but the rewards of the experts are difficult to summarize. Our key contribution here is to propose the MIMIC-MD algorithm which breaks the lower bound in Theorem (informal) 1.1 and suppresses the issue of error compounding which the covariate shift problem entails. Recent works [BSH20] propose algorithms that claim to bypass the covariate shift problem. However to the best of our knowledge, this is the first result that provably does so *in the general tabular MDP setting without additional assumptions*.

**Theorem (informal) 1.4** (Formal version: Theorem 4.1 (a)). *In the known-transition setting, if the expert is deterministic, the expected suboptimality of a learner $\widehat{\pi}$ playing MIMIC-MD is bounded by $J(\pi^*) - \mathbb{E}[J(\widehat{\pi})] \lesssim \min\{H\sqrt{|\mathcal{S}|/N}, |\mathcal{S}|H^{3/2}/N\}$.*

The novel element of MIMIC-MD is a hybrid approach which mimics the expert on some states, and uses a minimum distance (MD) functional [Yat85, DL88] to learn a policy on the remaining states. The minimum distance functional approach was recently considered in [SVBB19], proposing to sequentially learn a policy by approximately minimizing a notion of discrepancy between the learner's state distribution and the expert's empirical state distribution. We remark that our approach is fundamentally different from matching the state distributions under the expert and learner policy: it crucially relies on exactly mimicking the expert actions on states visited in the dataset, and only applying the MD functional on the remaining states.

Next we establish a lower bound on the error of any algorithm in the known-transition setting.

**Theorem (informal) 1.5** (Formal version: Theorem 5.1 (b)). *In the known-transition setting, for any learner $\widehat{\pi}$, there exists an instance such that the expected suboptimality $J(\pi^*) - \mathbb{E}[J(\widehat{\pi})] \gtrsim |\mathcal{S}|H/N$. This result applies even if the expert is constrained to be deterministic.*

Note that our suboptimality lower bounds corresponding to the no-interaction, active and known-transition settings (Informal Theorems 1.1 and 1.5) are universal and apply for any learner's policy $\widehat{\pi}$. In contrast, the lower bound example in [RB10] constructs an MDP showing that there exists a

particular learner policy which plays an action different than the expert with probability $\epsilon$ and has suboptimality $\gtrsim H^2\epsilon$. However, it turns out that the suboptimality incurred by behavior cloning is exactly 0 on the example provided in [RB10], *given just a single expert trajectory*.

## 1.2 Organization

In Section 1.3 we review related literature. In Section 2, we introduce necessary notations and definitions. In Section 3 we discuss the performance of behavior cloning and MIMIC-EMP in the no-interaction setting. In Section 4 we introduce and discuss our novel algorithm MIMIC-MD in the known transition setting. In Section 5 we complement the discussion in the previous sections with lower bounds. Technical proofs of our results are deferred to the supplementary material.

## 1.3 Related Work

The classical approach to IL focuses on learning from fixed expert demonstrations, e.g., [AN04, SBS08, RBZ06, ZMBD08, FLA16, HE16, PCS+17]. The reduction approach has also received much attention for theoretical analysis of IL [RB10, BSH20]. In the active setting, [RGB11] propose DAGGER, [RB14] propose AGGREVATE, and [SVG+17] propose AggreVaTeD which learn policies by actively interact with the environment and the expert during training. [LXM20] propsose a value function approach that is able to self-correct in IL. IL has also received attention from the general approach of minimizing f-divergences [KBS+19]. Very recently, [ADK+20] studies the imitation learning problem using a representation learning approach, where multiple agents' datasets are available for learning a common representation of the environment. While our results mainly focus on the case where both expert states and actions are observable, there are approaches e.g. [NCA+17, TWS18, SVBB19, ADK+20], studying the setting with observations of states alone. The statistical limits of IL in this setting is an interesting direction and is left as future work.

## 2 Preliminaries

An MDP $\mathcal{M} = (\mathcal{S}, \mathcal{A}, \rho, P, \mathbf{r}, H)$ describes the decision problem over state space $\mathcal{S}$ and action space $\mathcal{A}$. The initial state $s_1$ is drawn from a distribution $\rho$, and the state evolution at each time $t > 1$ is specified by unknown transition functions, $P = \{P_t(\cdot|s,a) : (s,a) \in \mathcal{S} \times \mathcal{A}\}_{t=1}^H$.

In addition, there is an unknown reward function $\mathbf{r} = (\mathbf{r}_1, \cdots, \mathbf{r}_H)$ where each $\mathbf{r}_t : \mathcal{S} \times \mathcal{A} \to [0,1]$. Choosing the action $a$ at state $s$ at time $t$, returns the reward $\mathbf{r}_t(s,a)$. Interaction with the MDP happens by rolling out a policy $\pi$, which is a non-stationary mapping from states to distributions over actions. Namely, $\pi = (\pi_1, \cdots, \pi_H)$ where $\pi_t : \mathcal{S} \to \Delta_1(\mathcal{A})$ and $\Delta_1(\mathcal{A})$ is the probability simplex over $\mathcal{A}$. We operate in the episodic setting and recall that the value of a policy $\pi$, defined in eq. (1), is the expected cumulative reward collected over an episode of length $H$. Here the $\mathbb{E}_\pi[\cdot]$ operator (resp. $\text{Pr}_\pi[\cdot]$) defines expectation (resp. probability) computed with respect to the trajectory generated by rolling out $\pi$, namely $\{s_1 \sim \rho; \forall t \in [H], a_t \sim \pi_t(\cdot|s_t),\ s_{t+1} \sim P_t(\cdot|s_t, a_t)\}$. Sometimes we instead use $J_\mathcal{M}(\cdot)$ or $J_\mathbf{r}(\cdot)$ in order to make the underlying MDP or reward function explicit. The suboptimality, $J(\pi^*) - J(\widehat{\pi})$, is the difference in value of the expert ($\pi^*$) and learner ($\widehat{\pi}$) policies.

Starting from the inital state $s_1 \sim \rho$, the learner *interacts* with the MDP by sequentially choosing actions $a_t$ at visited states $s_t$, with the MDP transitioning the learner to the next state $s_{t+1}$ sampled from $P_t(\cdot|s_t, a_t)$. In the imitation learning framework, the reward function $\mathbf{r}_t(s_t, a_t)$ is unobserved at each time $t$ in an episode. However, the learner can access demonstrations from an expert $\pi^*$ with the objective of learning a policy that has value comparable to the expert. We study imitation learning in the following 3 settings:

(a) **No-interaction:** The learner is provided a dataset $D$ of $N$ independent rollouts of the expert policy. The learner is not otherwise allowed to interact with the MDP.

(b) **Known-transition:** As in the no-interaction setting, the learner is provided an expert dataset $D$ of $N$ independent roll outs of the expert's policy. The learner additionally knows the MDP state transition functions $P$, as well as the initial distribution over states $\rho$.

(c) **Active:** The learner is not given a dataset of expert demonstrations in advance; however is allowed to interact with the MDP for $N$ episodes and can access an oracle, which upon being queried, returns the expert's action distribution $\pi_t^*(\cdot|s)$ at the learner's current state $s$.

In addition, we use the notation $\Pi_{\mathrm{det}}$ to denote the family of all deterministic policies. The notation $\pi_t^*(s)$ denotes the action played by a deterministic expert at state $s$ at time $t$.

# 3   No-interaction setting

We first study the setting where the expert policy $\pi^*$ is deterministic. The empirical 0-1 risk is the empirical frequency of the learner choosing an action different from the expert:

$$\mathbb{I}_{\mathrm{emp}}(\widehat{\pi}, \pi^*) = \frac{1}{H} \sum_{t=1}^{H} \mathbb{E}_{s_t \sim f_D^t}\left[\mathbb{E}_{a \sim \widehat{\pi}_t(\cdot|s_t)}\left[\mathbb{1}(a \neq \pi_t^*(s_t))\right]\right]. \tag{2}$$

Here $f_D^t$ is the empirical distribution over states at time $t$ averaged across trajectories in $D$. Analogously, we define the population 0-1 risk as,

$$\mathbb{I}_{\mathrm{pop}}(\widehat{\pi}, \pi^*) = \frac{1}{H} \sum_{t=1}^{H} \mathbb{E}_{s_t \sim f_{\pi^*}^t}\left[\mathbb{E}_{a \sim \widehat{\pi}_t(\cdot|s_t)}\left[\mathbb{1}(a \neq \pi_t^*(s_t))\right]\right]. \tag{3}$$

where $f_{\pi^*}^t$ is the distribution over states at time $t$ induced by rolling out the expert's policy $\pi^*$. Note that a policy that carries out behavior cloning and minimizes the empirical 0-1 risk to 0 in fact mimics the expert at all states observed in $D$. Since the policy on the remaining states is not specified, we define $\Pi_{\mathrm{mimic}}(D)$ as the set of all candidate deterministic policies that carry out behavior cloning,

$$\Pi_{\mathrm{mimic}}(D) \triangleq \left\{ \pi \in \Pi_{\mathrm{det}} : \forall t \in [H], s \in \mathcal{S}_t(D),\ \pi_t(\cdot|s) = \delta_{\pi_t^*(s)} \right\}, \tag{4}$$

where $\mathcal{S}_t(D)$ denotes the set of states visited at time $t$ in some trajectory in $D$ Indeed, the reduction approach in [RB10] shows that any policy $\widehat{\pi}$ that minimizes the population 0-1 loss to be $\leq \epsilon$ ensures that $J(\pi^*) - J(\widehat{\pi}) \leq H^2\epsilon$. We first analyze behavior cloning which minimizes the empirical 0-1 risk, and establish a generalization bound for the expected population 0-1 risk. We defer detailed proofs of the results in this section to Appendix A.1.

**Lemma 3.1** (Population 0-1 risk of Behavior Cloning). *Consider the no-interaction setting, and assume the expert policy $\pi^*$ is deterministic. Consider any policy $\widehat{\pi} \in \Pi_{\mathrm{mimic}}(D)$ (defined in eq. (4)). Then, the expected population 0-1 risk of $\widehat{\pi}$ (defined in eq. (3)) is bounded by,*

$$\mathbb{E}\left[\mathbb{I}_{\mathrm{pop}}(\widehat{\pi}, \pi^*)\right] \lesssim \min\left\{1, |\mathcal{S}/N\right\}. \tag{5}$$

*Proof Sketch.* The bound on the population 0-1 risk of behavior cloning relies on the following observation: at each time $t$, the learner exactly mimics the expert on the states that were visited in the expert dataset at least once. Therefore the contribution to the population 0-1 risk only stems from states that were never visited at time $t$ in any trajectory in $D$. Finally we identify that for each $t$, the probability mass contributed by such states has expected value upper bounded by $|\mathcal{S}|/N$.   □

With this result, invoking [RB10, Theorem 2.1] immediately results in the upper bound on the expected suboptimality of a learner carrying out behavior cloning in Theorem 3.2 (a). We use a similar approach to establish a high probability bound on the population 0-1 risk of behavior cloning.

**Theorem 3.2** (Upper bounding suboptimality of Behavior Cloning). *Consider any policy $\widehat{\pi}$ which carries out behavior cloning (i.e. $\widehat{\pi} \in \Pi_{\mathrm{mimic}}(D)$).*

(a) *The expected suboptimality of $\widehat{\pi}$ is upper bounded by,*

$$J(\pi^*) - \mathbb{E}\left[J(\widehat{\pi})\right] \lesssim \min\left\{H, |\mathcal{S}|H^2/N\right\}. \tag{6}$$

(b) *For $\delta \in (0, \min\{1, H/10\}]$, with probability $1 - \delta$ the suboptimality of $\widehat{\pi}$ is bounded by,*

$$J(\pi^*) - J(\widehat{\pi}) \lesssim |\mathcal{S}|H^2/N + \sqrt{|\mathcal{S}|}H^2 \log(H/\delta)/N. \tag{7}$$

*Proof Sketch.* We utilize the key observation in the proof of Lemma 3.1: for each $t = 1, \cdots, H$, the contribution to the population 0-1 risk in eq. (3) stems only from states that were never visited at time $t$ in any trajectory in $D$. For each $t$, we show that the mass contributed by such states up to constants does not exceed $|\mathcal{S}|/N + \sqrt{|\mathcal{S}|}\log(H/\delta)/N$ with probability $\geq 1 - \delta/H$. Summing over $t = 1, \cdots, H$ results in an upper bound on the population 0-1 loss that holds with probability $\geq 1 - \delta$ (by the union bound). Invoking [RB10, Theorem 2.1] proves the high probability bound.   □

## 3.1 Stochastic expert

We now move to the case where the expert policy is general and could be stochastic. Here, we consider MIMIC-EMP (Algorithm 1) where the learner minimizes the empirical risk under log-loss. This approach in fact corresponds to the learner playing the expert's empirical action distribution at states observed in the expert dataset.

---

**Algorithm 1** MIMIC-EMP

---

1: **Input:** Expert dataset $D$
2: **for** $t = 1, 2, \cdots, H$ **do**
3:      **for** $s \in \mathcal{S}$ **do**
4:          **if** $s \in \mathcal{S}_t(D)$ **then**
5:              $\widehat{\pi}_t(\cdot|s) = \pi_t^D(\cdot|s)$.         $\triangleright \pi_t^D(\cdot|s)$ is the empirical estimate of $\pi_t^*(\cdot|s)$ in dataset $D$
6:          **else**
7:              $\widehat{\pi}_t(\cdot|s) = \text{Unif}(\mathcal{A})$.
8: **Return** $\widehat{\pi}$

---

**Theorem 3.3.** *In the no-interaction setting, the learner's policy $\widehat{\pi}$ returned by* MIMIC-EMP *(Algorithm 1) has expected suboptimality upper bounded by,*

$$J(\pi^*) - \mathbb{E}\left[J(\widehat{\pi})\right] \lesssim \min\left\{H, \ |\mathcal{S}|H^2 \log(N)/N\right\} \tag{8}$$

*for a general expert $\pi^*$ which could be stochastic.*

We defer the formal analysis of Theorem 3.3 to Appendix A.2 and discuss some intuitions for the result below. In contrast to the setting where the expert is determinstic, it is no longer true that the learner is error-free as long as all states visited are observed in the expert dataset. However, by virtue of playing an empirical estimate of the expert's policy at these states it is plausible the expected suboptimality of the learner is $0$. However, a proof of this claim is not straightforward since the empirical distribution played by the learner at different states is not independent across time as functions of the dataset $D$.

We circumvent this problem by constructing a coupling between the expert's and learner's policies. Under the coupling it turns out the expected suboptimality of the learner is in fact $0$ when the visited states are all observed in the dataset. The remaining task is to bound the probability that at some point in the episode the learner visits a state unobserved in the expert dataset. A careful analysis of this probability term shows that it is bounded by $\lesssim |\mathcal{S}|H \log(N)/N$ under the coupling.

## 4 Known-transition setting

We next study imitation learning in the known-transition model where the initial state distribution $\rho$ and transition functions $P$ are known to the learner. To indicate this, we denote the learner's policy by $\widehat{\pi}(D, P, \rho)$. In this setting, mimicking the expert on states where the expert's policy is known is still a good approach, since there is no contribution to the learner's suboptimality as long as the learner only visits such states in an episode. With the additional knowledge of $P$, however, the learner can potentially do better on states that are unobserved in the demonstrations, and *correct* its mistakes even after it takes a wrong action, to avoid error compounding.

**Theorem 4.1.** *Consider the learner's policy $\widehat{\pi}$ returned by* MIMIC-MD *(Algorithm 2). When the expert policy $\pi^*$ is deterministic, in the known-transition setting,*

*(a) The expected suboptimality of the learner is upper bounded by,*

$$J(\pi^*) - \mathbb{E}\left[J(\widehat{\pi}(D, P, \rho))\right] \lesssim \min\left\{H, \ \sqrt{|\mathcal{S}|H^2/N}, \ |\mathcal{S}|H^{3/2}/N\right\}. \tag{9}$$

*(b) For $\delta \in (0, \min\{1, H/5\})$, with probability $1 - \delta$, the suboptimality of the learner satisfies,*

$$J(\pi^*) - J(\widehat{\pi}) \lesssim |\mathcal{S}|H^{3/2}/N \left(1 + 3\log(2|\mathcal{S}|H/\delta)/\sqrt{|\mathcal{S}|}\right)^{1/2} \sqrt{\log(2|\mathcal{S}|H/\delta)}. \tag{10}$$

---

**Algorithm 2** MIMIC-MD

---

1: **Input:** Expert dataset $D$.
2: Choose a uniformly random permutation of $D$,
   Define $D_1$ to be the first $N/2$ trajectories of $D$ and $D_2 = D \setminus D_1$.
3: Define $\mathcal{T}_t^{D_1}(s, a) \triangleq \{\{(s_{t'}, a_{t'})\}_{t'=1}^H | s_t = s, a_t = a, \exists \tau \leq t : s_\tau \notin \mathcal{S}_\tau(D_1)\}$ as trajectories that
   visit $(s, a)$ at time $t$, and at some time $\tau \leq t$ visit a state unvisited at time $\tau$ in any trajectory in $D_1$.
4: Define the optimization problem OPT below and return $\widehat{\pi}$ as any optimizer of it:

$$\min_{\pi \in \Pi_{\mathrm{mimic}}(D_1)} \sum_{t=1}^H \sum_{(s,a) \in \mathcal{S} \times \mathcal{A}} \left| \mathrm{Pr}_\pi\left[\mathcal{T}_t^{D_1}(s, a)\right] - \frac{1}{|D_2|} \sum_{\mathrm{tr} \in D_2} \mathbb{1}\left(\mathrm{tr} \in \mathcal{T}_t^{D_1}(s, a)\right) \right|. \quad \text{(OPT)}$$

      ▷ $\Pi_{\mathrm{mimic}}(D_1)$ is the set of policies that mimics the expert on the states visited in $D_1$ (eq. (4))
5: **Return** $\widehat{\pi}$

---

Theorem 4.1 (a) shows that MIMIC-MD (Algorithm 2) breaks the $|\mathcal{S}|H^2/N$ error compounding barrier which is not possible in the no-interaction setting, as discussed later in Theorem 5.1 (a). We defer the formal analysis of Theorem 4.1 to Appendix A.3 and provide intuitions for the result below.

MIMIC-MD inherits the spirit of mimicking the expert by exactly copying the expert actions in dataset $D_1$: as a result, the learner only incurs suboptimality upon visiting a state unobserved in $D_1$ at some point in an episode. Let $\mathcal{E}_{D_1}^{\leq t}$ be the event that the learner visits a state at some time $\tau \leq t$ which has not been visited in any trajectory in $D_1$ at time $\tau$. In particular, for any policy $\widehat{\pi}$ which mimics the expert on $D_1$, we show,

$$J(\pi^*) - J(\widehat{\pi}) \leq \sum_{s \in \mathcal{S}} \sum_{a \in \mathcal{A}} \sum_{t=1}^H \left| \mathrm{Pr}_{\pi^*}\left[\mathcal{E}_{D_1}^{\leq t}, s_t = s, a_t = a\right] - \mathrm{Pr}_{\widehat{\pi}}\left[\mathcal{E}_{D_1}^{\leq t}, s_t = s, a_t = a\right] \right|. \quad (11)$$

In the known-transition setting the learner knows the transition functions $\{P_t : 1 \leq t \leq H\}$ and the initial state distribution $\rho$, and can exactly compute the probability $\mathrm{Pr}_\pi[\mathcal{E}_{D_1}^{\leq t}, s_t = s, a_t = a]$ for any known policy $\pi$. However, unfortunately the learner cannot compute $\mathrm{Pr}_{\pi^*}[\mathcal{E}_{D_1}^{\leq t}, s_t = s, a_t = a]$ given only $D_1$. This is because the expert's policy on states unobserved in $D_1$ is unknown and the event $\mathcal{E}_{D_1}$ ensures that such states are necessarily visited. Here we use the remaining trajectories in the dataset, $D_2$ to compute an empirical estimate of $\mathrm{Pr}_{\pi^*}[\mathcal{E}_{D_1}^{\leq t}, s_t = s, a_t = a]$. The form of eq. (11) exactly motivates Algorithm 2, which replaces the population term $\mathrm{Pr}_{\pi^*}[\mathcal{E}_{D_1}^{\leq t}, s_t = s, a_t = a]$ by its empirical estimate in the MD functional.

**Remark 4.1.** *In the known-transition setting, the maximum likelihood estimate (MLE) for $\pi^*$ does not achieve the optimal sample complexity. When the expert is deterministic, all policies in $\Pi_{\mathrm{mimic}}(D)$ have equal likelihood given $D$. This is because the probability of observing a trajectory does not depend on the expert's policy on the states it does not visit. From Theorem 3.2 (a) and Item 5.1 (a) the expected suboptimality of the worst policy in $\Pi_{\mathrm{mimic}}(D)$ is $\asymp |\mathcal{S}|H^2/N$. Since the MLE does not give a rule to break ties, this implies that it is not optimal.*

**Remark 4.2.** *We conjecture that the conventional minimum distance functional approach, which matches the empirical distribution of either states or state-action pairs does not achieve the rate in Theorem 4.1 (a). Conventional distribution matching approaches do not take into account the fact that the expert's action is known at states visited in the dataset and may choose to play a different action at a state, even if the expert's action was seen in the dataset. In contrast, MIMIC-MD returns a policy that mimics the expert at states visited in the expert dataset, avoiding this issue.*

We also provide a guarantee when the learner returns any policy which solves the optimization problem OPT approximately to an additive accuracy of $\varepsilon$.

**Theorem 4.2.** *Consider any policy $\widehat{\pi}$ that minimizes the optimization problem OPT to an additive error of $\varepsilon$. Then, the expected suboptimality of the learner is upper bounded by,*

$$J(\pi^*) - \mathbb{E}\left[J(\widehat{\pi}(D, P))\right] \lesssim \min\left\{H, \ H\sqrt{|\mathcal{S}|/N} + \varepsilon, \ |\mathcal{S}|H^{3/2}/N + \varepsilon\right\}. \quad (12)$$

**Remark 4.3.** *As stated, it is not clear that MIMIC-MD is an efficient algorithm as it involves constrained optimization over high degree polynomials (in the learner's policy). However, Theorem 4.2 shows that it suffices to approximately minimize OPT to return a policy with small suboptimality. This can indeed be carried out efficiently and will be reported in a future work.*

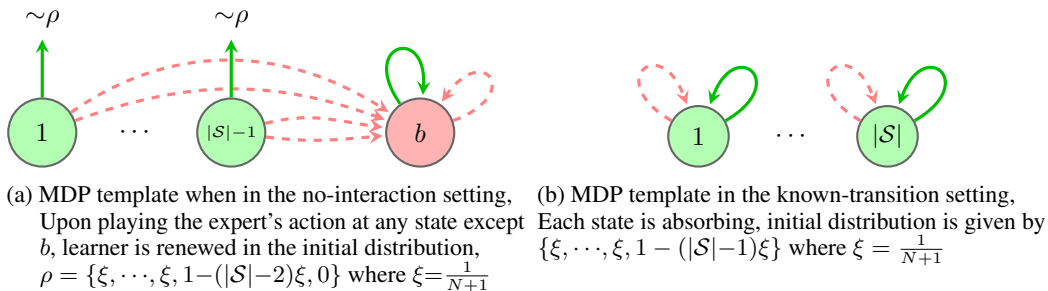

(a) MDP template when in the no-interaction setting, Upon playing the expert's action at any state except $b$, learner is renewed in the initial distribution, $\rho = \{\xi, \cdots, \xi, 1-(|\mathcal{S}|-2)\xi, 0\}$ where $\xi = \frac{1}{N+1}$

(b) MDP template in the known-transition setting, Each state is absorbing, initial distribution is given by $\{\xi, \cdots, \xi, 1-(|\mathcal{S}|-1)\xi\}$ where $\xi = \frac{1}{N+1}$

Figure 1: MDP templates for lower bounds under different settings: green arrows indicate state transitions under the expert's action, red arrows indicate state transitions under other actions

## 5 Lower bounds

**Theorem 5.1.** *For any learner $\widehat{\pi}$, for each $|\mathcal{S}| \geq 3$ and $|\mathcal{A}| \geq 2$ there exists an MDP $\mathcal{M}$ and a deterministic expert policy $\pi^*$ such that the expected suboptimality of $\widehat{\pi}$ is lower bounded in the:*

    *(a) no-interaction and active settings by, $J_{\mathcal{M}}(\pi^*) - \mathbb{E}[J_{\mathcal{M}}(\widehat{\pi})] \gtrsim \min\{H, |\mathcal{S}|H^2/N\}$.*

    *(b) known-transition setting by, $J_{\mathcal{M}}(\pi^*) - \mathbb{E}[J_{\mathcal{M}}(\widehat{\pi})] \gtrsim \min\{H, |\mathcal{S}|H/N\}$.*

We construct the worst case MDP templates for the no-interaction and active settings in Figure 1a and that for the known-transition setting in Figure 1b and defer the formal analysis to Appendix A.4.

In Figure 1a, at any state of the MDP, except one, every other action, moves the learner to the absorbing state $b$. Suppose a learner independently plays an action different from the expert at a state with probability $\epsilon$. Upon making a mistake, the learner is transferred to $b$ and collects no reward for the rest of the episode. Thus the suboptimality of the learner is $\geq H\epsilon + (H-1)\epsilon(1-\epsilon) + \cdots + (1-\epsilon)^H \gtrsim \min\{H, H^2\epsilon\}$. By construction of $\rho$, we identify that any learner must make a mistake with probability $\epsilon \gtrsim |\mathcal{S}|/N$, resulting in the claim.

**Remark 5.1.** *The lower bound construction in Figure 1a applies even if the learner can actively query the expert while interacting with the MDP. If the expert's queried action is not followed at any state, the learner is transitioned to $b$ with probability $1$. Upon doing so, the learner no longer can get any meaningful information about the expert's policy at states for the rest of the episode. Seeing that the "most informative" dataset the learner can collect involves following the expert at each time, it is no different had an expert dataset of $N$ trajectories been provided in advance. This reduces the active case to the no-interaction case for which the existing construction applies.*

The lower bound construction in the known-transition setting is illustrated in Figure 1b: each state in the MDP is absorbing so a policy stays at a single state in an episode. If the initial state of the MDP was not visited in the dataset, the learner does not see the expert's action for the rest of the episode which is the only one to offer non-zero reward: at such states the learner's suboptimality is $\gtrsim H$. By construction of $\rho$, the probability of initialization in such a state is $\gtrsim |\mathcal{S}|/N$, resulting in the claim.

## 6 Conclusion

We show that behavior cloning is in fact optimal in the no-interaction setting, when the expert is deterministic. In addition, we show that minimizing empirical risk under log-loss results in a policy which is optimal up to logarithmic factors even when the expert is stochastic. In the known-transition setting we propose the first policy that provably breaks the $H^2$ error compounding barrier, and show a lower bound which it matches up to a $\sqrt{H}$-factor. An important question we raise is to bridge this gap between the upper and lower bounds in the known-transition setting. We study IL at two opposing ends of the spectrum: when the learner cannot interact with the MDP, and when the learner exactly knows the transition structure. It is a fundamental question to ask is how much improvement is possible when the learner is allowed to interact with the MDP a finite number of times. It is also an interesting question to extend these results beyond the tabular setting.

## Broader Impact

An important conclusion of our work is that algorithms that require an expert that can be actively queried, in the worst case, do not break the error compounding barrier. While it is plausible that such algorithms do indeed perform better in practical problems, it still raises an important point that compounding errors is fundamental to IL and requires a better understanding in practice. Furthermore, our proposed algorithm MIMIC-MD in the known-transition setting reveals some new algorithmic primitives that can hopefully inspire practitioners to design algorithms that perform better.

## Acknowledgments and Disclosure of Funding

This work was partially supported by NSF Grants IIS-1901252, CCF-1909499, and CIF-1703678.

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
