[Supplementary Material]

# Supplement for "Toward the Fundamental Limits of Imitation Learning"

## Appendix A

## Contents

We provide proofs for the theorems introduced previously in this appendix. We push the proofs of some of the lemmas and claims invoked in this section to Appendix B. For the remainder of the paper, we use $\log(\cdot)$ to denote the natural logarithm.

### A.1 No-interaction setting under deterministic expert policy

In this section, we discuss the no-interaction setting where the learner is provided access to a dataset $D$ of $N$ trajectories generated by rolling out the expert's policy $\pi^*$, and is otherwise not allowed to interact with the MDP. Our goal is to provide guarantees on the expected suboptimality incurred by a policy that carries out behavior cloning when the expert's policy is deterministic. As stated previously, we realize this guarantee by first bounding the population 0-1 risk of behavior cloning (Theorem 3.1) and then invoking the black box reduction guarantee from [RB10].

#### A.1.1 Analysis of expected suboptimality of behavior cloning

We first discuss the proof of Theorems 3.1 and 3.2 (a), which bound the expected suboptimality of a policy carrying out behavior cloning, assuming the expert's policy is deterministic.

Recall that the population 0-1 loss is defined as,

$$\mathbb{I}_{\text{pop}}(\widehat{\pi}, \pi^*) = \frac{1}{H} \sum_{t=1}^{H} \mathbb{E}_{s_t \sim f_{\pi^*}^t} \left[ \mathbb{E}_{a \sim \widehat{\pi}_t(\cdot|s_t)} \left[ \mathbb{1}(a \neq \pi_t^*(s)) \right] \right] \tag{13}$$

where $f_{\pi^*}^t$ is the state distribution induced at time $t$ rolling out the expert's policy $\pi^*$. We consider a learner $\widehat{\pi}$ that carries out behavior cloning given the expert dataset $D$ in advance. In particular, the learner's policy $\widehat{\pi}$ is a member of $\Pi_{\text{mimic}}(D)$ since it exactly mimics the expert on the states that were visited at each time in some trajectory in the expert dataset. Thus the contribution to the population 0-1 risk comes from the remaining states $s \in \mathcal{S}_t(D)$,

$$\mathbb{I}_{\text{pop}}(\widehat{\pi}, \pi^*) \leq \frac{1}{H} \sum_{t=1}^{H} \mathbb{E}_{s_t \sim f_{\pi^*}^t} \left[ \mathbb{1}(s_t \notin \mathcal{S}_t(D)) \right] \tag{14}$$

$$= \frac{1}{H} \sum_{t=1}^{H} \sum_{s \in \mathcal{S}} \Pr_{\pi^*}[s_t = s] \mathbb{1}(s \notin \mathcal{S}_t(D)). \tag{15}$$

Taking expectation on both sides gives,

$$\mathbb{E}\left[\mathbb{l}_{\mathrm{pop}}(\widehat{\pi}, \pi^*)\right] \leq \frac{1}{H}\sum_{t=1}^{H}\sum_{s\in\mathcal{S}} \mathrm{Pr}_{\pi^*}[s_t = s]\mathrm{Pr}(s \notin \mathcal{S}_t(D)). \tag{16}$$

In Lemma A.1 we show that this expression is bounded by $\lesssim |\mathcal{S}|H/N$, which completes the proof of the population 0-1 risk bound of behavior cloning in Theorem 3.1.

**Lemma A.1.** $\mathbb{E}\left[\sum_{t=1}^{H}\sum_{s\in\mathcal{S}}\mathrm{Pr}_{\pi^*}[s_t = s]\,\mathrm{Pr}[s \notin \mathcal{S}_t(D)]\right] \leq \frac{4}{9}\frac{|\mathcal{S}|H}{|D|}$.

As stated previously, we subsequently invoke the supervised learning reduction in [RB10, Theorem 2.1] to provide the guarantee on the expected suboptimality of behavior cloning in Theorem 3.2 (a).

The previous discussion is also amenable for establishing a high probability bound on the expected suboptimality of behavior cloning. Indeed, consider the upper bound on the population 0-1 risk of behavior cloning in eq. (15), which is a function of the expert dataset $D$. It captures the probability mass under the expert's state distribution contributed by states unobserved in the expert dataset.

### A.1.2 High probability bounds for behavior cloning

It turns out that the contribution to the upper bound on population 0-1 risk of behavior cloning in eq. (15) is captured by the notion of "missing mass" of the time-averaged state distribution under the expert's policy. The high probability result Theorem 3.2 (b) for behavior cloning follows shortly by invoking existing concentration bounds for missing mass.

**Definition A.1** (Missing mass). *Consider some distribution $\nu$ on $\mathcal{X}$, and let $X^N \overset{i.i.d.}{\sim} \nu$ be a dataset of $N$ samples drawn i.i.d. from $\nu$. Let $\mathfrak{n}_x(X^N) = \sum_{i=1}^{N}\mathbb{1}(X_i = x)$ be the number of times the symbol $x$ was observed in $X^N$. Then, the missing mass $\mathfrak{m}_0(\nu, X^N) = \sum_{x\in\mathcal{X}}\nu(x)\mathbb{1}(\mathfrak{n}_x = 0)$ is the probability mass contributed by symbols never observed in $X^N$.*

It turns out that the missing mass of discrete distributions admit tight concentration about its mean.

**Theorem A.2.** *Consider an arbitrary distribution $\nu$ on $\mathcal{X}$, and let $X^N \overset{i.i.d.}{\sim} \nu$ be a dataset of $N$ samples drawn i.i.d. from $\nu$. Consider any $\delta \in (0, 1/10]$. Then,*

$$\mathrm{Pr}\left(\mathfrak{m}_0(\nu, X^N) - \mathbb{E}[\mathfrak{m}_0(\nu, X^N)] \geq \frac{3\sqrt{|\mathcal{X}|}\log(1/\delta)}{N}\right) \leq \delta. \tag{17}$$

Consider the upper bound to the population 0-1 loss in eq. (16). Observe that for each fixed $\tau \in [H]$, $\sum_{s\in\mathcal{S}}\mathrm{Pr}_{\pi^*}[s_\tau = s]\mathbb{1}(s \notin \mathcal{S}_\tau(D))$ is the missing mass of $f_{\pi^*}^\tau$, given $N$ samples from the distribution. Recall that $f_{\pi^*}^\tau$ is the distribution over states at time $\tau$ rolling out $\pi^*$. Thus we can invoke the concentration bound from Theorem A.2 to prove that the upper bound on 0-1 loss in eq. (16) concentrates. We formally state this result in Lemma A.3.

**Lemma A.3.** *For any $\delta$ such that $\delta \in (0, \min\{1, H/10\}]$, with probability $\geq 1 - \delta$ over the randomness of the expert dataset $D$,*

$$\frac{1}{H}\sum_{\tau=1}^{H}\sum_{s\in\mathcal{S}}\mathrm{Pr}_{\pi^*}[s_\tau = s]\mathbb{1}(s \notin \mathcal{S}_\tau(D)) \leq \frac{4|\mathcal{S}|}{9N} + \frac{3\sqrt{|\mathcal{S}|}\log(H/\delta)}{N}. \tag{18}$$

Plugging this result into eq. (16) provides an upper bound on the population 0-1 risk of behavior cloning. Subsequently invoking [RB10, Theorem 2.1], we arrive at the high probability bound on $J(\pi^*) - J(\widehat{\pi})$ for behavior cloning in Theorem 3.2 (b).

### A.2 No-interaction setting when the expert policy is stochastic

In this section we continue to discuss the no-interaction setting, but drop the assumption that the expert plays a deterministic policy. We assume the expert plays a general stochastic policy. Following the reduction approach in [RB10], we first show a supervised learning reduction from imitation learning to matching the expert's policy in total variation (TV) distance. The ensuing discussion however shows that the reduction analysis is loose when the expert policy is stochastic.

### A.2.1 Reduction of IL to supervised learning under TV distance

[RB10, Theorem 2.1] show that if the expert's policy is deterministic, and the probability of guessing the expert's action at each state is $\epsilon$, then $J(\pi^*) - J(\widehat{\pi}) \leq \min\{H, \epsilon H^2\}$. In this section we prove a generalization of this result which applies even if the expert plays a stochastic policy. To this end, we first introduce the population TV risk,

$$\mathbb{T}_{\mathrm{pop}}(\widehat{\pi}, \pi^*) = \frac{1}{H} \sum_{t=1}^{H} \mathbb{E}_{s_t \sim f_{\pi^*}^t} \left[ \mathsf{TV}\left( \widehat{\pi}_t(\cdot|s_t), \pi_t^*(\cdot|s_t) \right) \right]. \tag{19}$$

We show that if the learner minimizes the population TV risk to be $\leq \epsilon$ then the expected suboptimality of the learner is $\lesssim \min\{H, H^2\epsilon\}$. The population TV risk of a learner is a generalization of the population 0-1 risk to the case where the expert's policy is stochastic. We formally state the reduction below.

**Lemma A.4.** *Consider any policy $\widehat{\pi}$ such that $\mathbb{T}_{\mathrm{pop}}(\widehat{\pi}, \pi^*) \leq \epsilon$. Then, $J(\pi^*) - J(\widehat{\pi}) \leq \min\{H, H^2\epsilon\}$.*

**Remark A.1.** *When the expert is deterministic, the definition of $\mathbb{T}_{\mathrm{pop}}$ matches that of $\mathbb{I}_{\mathrm{pop}}$ (Lemma 3.1) recovering the guarantee in [RB10, Theorem 2.1]. Thus, Lemma A.4 strictly generalizes the supervised learning reduction for behavior cloning.*

While the reduction approach seems promising at first, there is a catch - the population TV risk in fact converges very slowly to $0$. Since it corresponds to matching the expert's and learner's action distributions, the convergence rate is $\gtrsim \sqrt{|\mathcal{A}|/N}$ even if $|\mathcal{S}| = 1$. In the same setting, the population 0-1 risk which is the counterpart in the deterministic expert setting converges at a much faster $1/N$ rate (Theorem 3.1).

At first, the optimality of the reduction approach when the expert is deterministic seems to suggest that imitation learning may be a significantly harder problem when the expert is stochastic. However, we show that this is in fact not the case and circumvent the reduction framework to prove this result. We show in Theorem 3.3 that an expected suboptimality up to logarithmic factors achieving the same $1/N$ rate of convergence can be realized. We show that a natural policy MIMIC-EMP achieves this rate - one which plays the empirical estimate of the expert's policy wherever available, and the uniform distribution over actions otherwise. Moreover the guarantee on the expected suboptimality of this policy is optimal in the dependence on $H$ and achieves the compounding error lower bound in Theorem 5.1 (a).

### A.2.2 Analyzing expected suboptimality of MIMIC-EMP

In this section we discuss the proof of Theorem 3.3 which bounds the expected suboptimality incurred by MIMIC-EMP. Recall that the objective is to upper bound $J(\pi^*) - \mathbb{E}[J(\widehat{\pi})]$ when the learner carries out MIMIC-EMP. The outline of the proof is to construct two policies $\pi^{\mathrm{first}}$ and $\pi^{\mathrm{orc-first}}$ that are functions of the dataset $D$.

The policy $\pi^{\mathrm{first}}$ is easy to describe: order the expert dataset arbitrarily, and at a state, play the action in the first trajectory in $D$ that visits it, if it exists. If no such trajectories exist, the policy plays $\mathrm{Unif}(\mathcal{A})$. In particular, we show that the value of $\pi^{\mathrm{first}}$ and MIMIC-EMP are the same, taking expectation over the expert dataset $D$ (Lemma A.5).

On the other hand, we consider an oracle policy $\pi^{\mathrm{orc-first}}$ which is very similar. Indeed, $\pi^{\mathrm{orc-first}}$ first orders the expert dataset in the same manner as $\pi^{\mathrm{first}}$. At any state, it too plays the action in the first trajectory in $D$ that visits it, if it exists. However, if such a trajectory does not exist, $\pi^{\mathrm{orc-first}}$ simply samples an action from the expert's action distribution and plays it at this state. This explains the namesake of the policy, since it requires oracle access to the expert's policy. By virtue of choosing actions this way, we show that the value of $\pi^{\mathrm{orc-first}}$ in expectation equals $J(\pi^*)$ (Lemma A.7).

At an intuitive level the elements of the proof seem to be surfacing: $\pi^{\mathrm{orc-first}}$ matches $\pi^*$ in value, but is not available to the learner. However, it shares a lot of similarity to $\pi^{\mathrm{first}}$, which in expectation matches $\widehat{\pi}$ in value, the policy we wish to analyze. Informally,

$$\widehat{\pi} \iff \pi^{\mathrm{first}} \approx \pi^{\mathrm{orc-first}} \iff \pi^* \tag{20}$$

Thus to establish the bound, we carry out an analysis of $J(\pi^{\mathrm{orc-first}}) - J(\pi^{\mathrm{first}})$. Indeed we show that since the two policies are largely the same, an error is incurred only on the trajectories where at

some point a state is visited where the policies do not match. The final element of the proof is to show that this event in fact occurs with low probability given an expert dataset of sufficiently large size.

Before delving into the formal definitions of $\pi^{\text{first}}$ and $\pi^{\text{orc}-\text{first}}$ and other elements of the proof, we introduce a modicum of relevant notation.

**Notation:** We assume that the trajectories in the expert dataset $D$ are ordered arbitrarily as $\{\text{tr}_1, \cdots, \text{tr}_N\}$. In addition, we denote each trajectory $\text{tr}_n$ explicitly as $\{(s_1^n, a_1^n), \cdots, (s_H^n, a_H^n)\}$. For each state $s \in \mathcal{S}$ we define,

$$N_{t,s} = \{n \in [N] : s_t^n = s\}, \tag{21}$$

as the (totally) ordered set of indices of trajectories in $D$ which visit the state $s$ at time $t$.

The policy $\widehat{\pi}$ returned by MIMIC-EMP samples an action from the empirical estimate of the expert's policy at each state wherever available. On the remaining states, the learner plays the distribution $\text{Unif}(\mathcal{A})$.

Given the ordered dataset $D$, we define the policy $\pi^{\text{first}}(D)$ as,

$$\pi_t^{\text{first}}(\cdot|s) = \begin{cases} \delta_{a_t^n} & \text{if } |N_{t,s}| \geq 1, \text{ where } n = \min(N_{t,s}), \\ \text{Unif}(\mathcal{A}) & \text{otherwise.} \end{cases} \tag{22}$$

In other words, $\pi^{\text{first}}(D)$ plays the action in the first trajectory that visits the state $s$ at time $t$.

In order to analyze the expected suboptimality of $\widehat{\pi}(D)$, we first show that $\widehat{\pi}(D)$ and $\pi^{\text{first}}(D)$ have the same value in expectation, and instead study the policy $\pi^{\text{first}}(D)$.

**Lemma A.5.** $\mathbb{E}[J(\widehat{\pi}(D))] = \mathbb{E}[J(\pi^{\text{first}}(D))]$.

With this result, we can write the expected suboptimality of the learner $\widehat{\pi}$ as,

$$J(\pi^*) - \mathbb{E}[J(\widehat{\pi}(D))] = J(\pi^*) - \mathbb{E}[J(\pi^{\text{first}}(D))]. \tag{23}$$

We next move on to the discussion of $\pi^{\text{orc}-\text{first}}$ which is an oracle version of $\pi^{\text{first}}$. Informally, at any state $\pi^{\text{orc}-\text{first}}$ plays the action from the first trajectory that visits it in $D$, if available. However on the remaining states instead of playing $\text{Unif}(\mathcal{A})$, $\pi^{\text{orc}-\text{first}}$ samples an action from the expert's action distribution and plays it at this state. Thus, $\pi^{\text{orc}-\text{first}}$ is coupled with the expert dataset $D$.

Prior to discussing $\pi^{\text{orc}-\text{first}}$ in greater depth, we first introduce some preliminaries. In particular, we adopt an alternate view of the process generating the expert dataset $D$ which will play a central role in formally defining $\pi^{\text{orc}-\text{first}}$. We mention that this approach is inspired by the alternate view of Markov processes in [Bil61].

To this end, we first define an "expert table" which is a fixed infinite collection of actions at each state and time which the expert draws upon while generating the trajectories in $D$.

**Definition A.2** (Expert table). *The expert table, $\mathbf{T}^*$ is a collection of random variables $\mathbf{T}_{t,s}^*(i)$ indexed by $t \in [H]$, $s \in \mathcal{S}$ and $i = 1, 2, \cdots$. Fixing $s \in \mathcal{S}$ and $t \in [H]$, for $i = 1, 2, \cdots$, each $\mathbf{T}_{t,s}^\pi(i)$ is drawn independently $\sim \pi_t^*(\cdot|s)$.*

In a sense, the expert table fixes the randomness in the expert's non-determinstic policy. As promised, we next present the alternate view of generating the expert dataset $D$, where the expert sequentially samples actions from the expert table at visited states.

**Lemma A.6** (Alternate view of generating $D$). *Generate a dataset $D$ of $N$ trajectories as follows: For the $n^{th}$ trajectory $\text{tr}_n$, the state $s_1^n$ is drawn independently from $\rho$. The action $a_1^n$ is assigned as the first action from $\mathbf{T}_{1,s_1^n}^*(\cdot)$ that was not chosen in a previous trajectory. Then the MDP independently samples the state $s_2^n \sim P_1(\cdot|s_1^n, a_1^n)$. In general, at time $t$ the action $a_t^n$ is drawn as the first action in $\mathbf{T}_{t,s_t^n}^*(\cdot)$ that was not chosen at time $t$ in any previous trajectory $n' < n$. The subsequent state $s_{t+1}^n$ is drawn independently $\sim P_{t+1}(\cdot|s_t^n, a_t^n)$.*

*The probability of generating a dataset $D = \{\text{tr}_1, \cdots, \text{tr}_N\}$ by this procedure is $= \prod_{n=1}^N \Pr_{\pi^*}[\text{tr}_n]$. This is the same as if the trajectories were generated by independently rolling out $\pi^*$ for $N$ episodes.*

*Proof.* Starting from the initial state $s_1^n \sim \rho$, the probability of $\text{tr}_n = \{(s_1, a_1), \cdots, (s_H, a_H)\}$ is,

$$\Pr\Big(\text{tr}_n = \{(s_1, a_1), \cdots, (s_H, a_H)\}\Big) = \rho(s_1) \Big(\prod_{t < H} \pi_t^*(a_t|s_t) P_t(s_{t+1}|s_t, a_t)\Big) \pi_H^*(a_H|s_H).$$

This relies on the fact that each action in $\mathbf{T}_{t,s}^*(\cdot)$ is sampled independently from $\pi_t^*(\cdot|s)$. Carrying out the same calculation for the $n$ trajectories jointly (which we avoid to keep notation simple) results in the claim. The important element remains the same: each action in $\mathbf{T}_{t,s_t^n}^*(\cdot)$ is sampled independently from $\pi_t^*(\cdot|s_t^n)$. $\qquad \square$

Note that the process in Lemma A.6 generates a dataset having the same distribution as if the trajectories were generated by independently rolling out $\pi^*$ for $N$ episodes. Without loss of generality we may therefore assume that the expert generates $D$ this way. We adopt this alternate view to enable the coupling between the expert's and learner's policies.

**Remark A.2.** *We emphasize that the infinite table $\mathbf{T}^*$ is not known to the learner and is only used by the expert to generate the dataset $D$. However, by virtue of observing the trajectories in $D$ the learner is revealed some part of $\mathbf{T}^*$. In particular at the state $s$ and time $t$, the first $|N_{t,s}|$ actions in $\mathbf{T}_{t,s}^*$ are revealed to the learner.*

Recall that $\pi_t^{\text{first}}(\cdot|s)$ defined in eq. (22) deterministically plays the action in the first trajectory in $D$ that visits a state $s$ at time $t$, if available, and otherwise plays the uniform distribution $\text{Unif}(\mathcal{A})$.

Using the alternate view of generating $D$ in Lemma A.6, this policy can be equivalently defined as one which plays the action at the first position in the table $\mathbf{T}^*$ if observed, and otherwise plays the uniform distribution.

$$\pi_t^{\text{first}}(\cdot|s) = \begin{cases} \delta_{\mathbf{T}_{t,s}^*(1)} & \text{if } |N_{t,s}| > 0, \\ \text{Unif}(\mathcal{A}), & \text{otherwise.} \end{cases} \tag{24}$$

We now define the oracle policy $\pi^{\text{orc}-\text{first}}$, which plays the first action at each time $t \in [H]$ at each state $s \in \mathcal{S}$. That is,

$$\pi_t^{\text{orc}-\text{first}}(\cdot|s) = \delta_{\mathbf{T}_{t,s}^*(1)} \tag{25}$$

With this definition, we first identify that the expected value of $\pi^{\text{orc}-\text{first}}$ equals $J(\pi^*)$.

**Lemma A.7.** $J(\pi^*) = \mathbb{E}\left[J(\pi^{\text{orc}-\text{first}})\right].$

Plugging this into eq. (23), we see that,

$$J(\pi^*) - \mathbb{E}[J(\widehat{\pi}(D))] = \mathbb{E}\left[J(\pi^{\text{orc}-\text{first}}) - J(\pi^{\text{first}})\right]. \tag{26}$$

Observe that $\pi^{\text{orc}-\text{first}}$ and $\pi^{\text{first}}$ are in fact identical on all the states that were visited at least once in the expert dataset (i.e. having $|N_{t,s}| > 0$). Therefore, as long as the state $s$ visited at each time $t$ in an episode has $|N_{t,s}| > 0$, both policies collect the same cumulative reward.

**Lemma A.8.** *Fix the expert table $\mathbf{T}^*$ and the expert dataset $D$. Define $\mathcal{E}^c$ as the "good" event that the trajectory under consideration only visits a state $s_t$ at each time $t \in [H]$ such that $|N_{t,s_t}| > 0$, i.e. states that have been observed in the expert dataset $D$ at time $t$. Then,*

$$\mathbb{E}_{\pi^{\text{first}}}\left[\left(\sum_{t=1}^{H} \mathbf{r}_t(s_t, a_t)\right) \mathbb{1}(\mathcal{E}^c)\right] = \mathbb{E}_{\pi^{\text{orc}-\text{first}}}\left[\left(\sum_{t=1}^{H} \mathbf{r}_t(s_t, a_t)\right) \mathbb{1}(\mathcal{E}^c)\right] \tag{27}$$

*Proof.* Both policies are identical on the states such that $|N_{t,s}| > 0$. The event $\mathcal{E}^c$ guarantees that only such states are visited in a trajectory. Therefore both expectations are equal. $\qquad \square$

With these preliminaries, we have most of the ingredients to prove the bound on the expected suboptimality incurred by MIMIC-EMP. To this end, from eq. (26) we see that,

$$J(\pi^*) - J(\widehat{\pi}(D)) = \mathbb{E}_{\pi^{\text{orc}-\text{first}}}\left[\sum_{t=1}^{H} \mathbf{r}_t(s_t, a_t)\right] - \mathbb{E}_{\pi^{\text{first}}}\left[\sum_{t=1}^{H} \mathbf{r}_t(s_t, a_t)\right] \tag{28}$$

Subsequently invoking Lemma A.8, we see that

$$J(\pi^*) - J(\widehat{\pi}(D)) = \mathbb{E}_{\pi^{\text{orc-first}}}\left[\left(\sum_{t=1}^{H}\mathbf{r}_t(s_t,a_t)\right)\mathbb{1}\left(\mathcal{E}\right)\right] - \mathbb{E}_{\pi^{\text{first}}}\left[\left(\sum_{t=1}^{H}\mathbf{r}_t(s_t,a_t)\right)\mathbb{1}\left(\mathcal{E}\right)\right]$$

$$\leq \mathbb{E}_{\pi^{\text{orc-first}}}\left[\left(\sum_{t=1}^{H}\mathbf{r}_t(s_t,a_t)\right)\mathbb{1}\left(\mathcal{E}\right)\right] \tag{29}$$

$$\leq H\mathrm{Pr}_{\pi^{\text{orc-first}}}\left[\mathcal{E}\right] \tag{30}$$

where in the last inequality we use the fact that pointwise $0 \leq \mathbf{r}_t \leq 1$ for all $t \in [H]$. Taking expectation gives the inequality,

$$J(\pi^*) - \mathbb{E}[J(\widehat{\pi}(D))] \leq H\mathbb{E}[\mathrm{Pr}_{\pi^{\text{orc-first}}}\left[\mathcal{E}\right]] \tag{31}$$

In Lemma A.9 we show that $\mathbb{E}[\mathrm{Pr}_{\pi^{\text{orc-first}}}\left[\mathcal{E}\right]]$ is upper bounded by $|\mathcal{S}|H\ln(N)/N$, which completes the proof.

**Lemma A.9.** $\mathbb{E}\left[\mathrm{Pr}_{\pi^{\text{orc-first}}}[\mathcal{E}]\right] \leq \frac{|\mathcal{S}|H\ln(N)}{N}$.

Although the oracle policy $\pi^{\text{orc-first}}$ and the dataset $D$ are coupled, the key intuition behind showing that the event $\mathcal{E}$ occurs with low probability is that: it is not possible that, in expectation $\pi^{\text{orc-first}}$ visits some state $s$ with high probability, but the same state $s$ visited in the dataset $D$ with low probability. This is by virtue of the fact that in expectation $\pi^{\text{orc-first}}$ matches $\pi^*$ which is the policy that generates $D$.

## A.3 Known-transition setting under deterministic expert policy

In this section, we describe the proof of Theorem 4.1 (a) which upper bounds the expected suboptimality of MIMIC-MD (Algorithm 2).

Recall that MIMIC-MD, true to its name, mimics the expert on the states observed in half the dataset $D_1$. By virtue of the learner mimicking the expert on states visited in $D_1$, we show that the learner incurs error only upon visiting a state unobserved in $D_1$ at some point in an episode.

**Lemma A.10.** *Define* $\mathcal{E}_{D_1} = \{\exists t \in [H] : s_t \notin \mathcal{S}_t(D_1)\}$ *as the event that the policy under consideration visits some state at time* $t$ *that no trajectory in* $D_1$ *has visited at time* $t$.

*Fixing the expert datset $D$, for any policy $\widehat{\pi} \in \Pi_{\text{mimic}}(D_1)$,*

$$J(\pi^*) - J(\widehat{\pi}(D)) = \mathbb{E}_{\pi^*}\left[\mathbb{1}(\mathcal{E}_{D_1})\sum_{t=1}^{H}\mathbf{r}_t(s_t,a_t)\right] - \mathbb{E}_{\widehat{\pi}(D)}\left[\mathbb{1}(\mathcal{E}_{D_1})\sum_{t=1}^{H}\mathbf{r}_t(s_t,a_t)\right]. \tag{32}$$

Simplifying this result further using the fact that the reward function is bounded in $[0,1]$ results in eq. (11), recall which we used as a basis for motivating the design of MIMIC-MD in Section 4. In particular, any policy $\widehat{\pi}$ that exactly mimics the expert on states observed in $D_1$ has suboptimality bounded by,

$$J(\pi^*) - J(\widehat{\pi}) \leq \sum_{s\in\mathcal{S}}\sum_{a\in\mathcal{A}}\sum_{t=1}^{H}\left|\mathrm{Pr}_{\pi^*}\left[\mathcal{E}_{D_1}, s_t = s, a_t = a\right] - \mathrm{Pr}_{\widehat{\pi}}\left[\mathcal{E}_{D_1}, s_t = s, a_t = a\right]\right|$$

The minimum distance functional considered in MIMIC-MD simply replaces the population term $\mathrm{Pr}_{\pi^*}[\mathcal{E}_{D_1}, s_t = s, a_t = a]$ by its empirical estimate computed using the dataset $D_2$. We follow the standard analysis of minimum distance function estimators using the triangle inequality, which in effect reduces the analysis to a question of convergence of the empirical estimate of $\mathrm{Pr}_{\pi^*}[\mathcal{E}_{D_1}, s_t = \cdot, a_t = \cdot]$ to the population in $\ell_1$ distance.

Before stating the formal lemma we recall that

$$\mathcal{T}_t^{\mathcal{E}_{D_1}}(s,a) \triangleq \left\{\{(s_1,a_1),\cdots,(s_H,a_H)\}\Big|s_t = s, a_t = a, \exists\tau\in[H]: s_\tau \notin \mathcal{S}_\tau(D_1)\right\}. \tag{33}$$

is defined as the set of trajectories that (i) visits the state $s$ at time $t$, (ii) plays the action $a$ at this time, and (iii) at some time $\tau \in [H]$ visit a state unobserved in $D_1$.

**Lemma A.11.** *Consider any policy $\widehat{\pi}^\varepsilon \in \Pi_{\mathrm{mimic}}(D)$ which solves the optimization problem in* OPT *to an additive error of $\varepsilon$. Fixing the expert dataset $D$,*

$$J(\pi^*) - J(\widehat{\pi}^\varepsilon(D)) \le 2 \sum_{s\in\mathcal{S}} \sum_{a\in\mathcal{A}} \sum_{t=1}^{H} \left| \Pr_{\pi^*}\left[ \mathcal{E}_{D_1}^{\le t}, s_t = s, a_t = a \right] - \frac{\sum_{tr\in D_2} \mathbb{1}(tr \in \mathcal{T}_t^{D_1}(s,a))}{|D_2|} \right| + \varepsilon.$$

We emphasize here that $\frac{\sum_{tr\in D_2} \mathbb{1}(tr\in\mathcal{T}_t^{D_1}(s,a))}{|D_2|}$ is the empirical estimate of $\Pr_{\widehat{\pi}}\left[ \mathcal{E}_{D_1}^{\le t}, s_t = s, a_t = a \right]$ computed using the trajectories in the dataset $D_2$.

**Remark A.3.** *Taking $\varepsilon = 0$ in Lemma A.11 captures the case where $\widehat{\pi}^\varepsilon$ is the policy returned by* MIMIC-MD.

The last remaining ingredient in proving the expected suboptimality guarantee of MIMIC-MD in Theorem 4.1 (a) is to bound the convergence rate of the expectation of the RHS of Lemma A.11. We carry out this analysis roughly in two parts:

(i) fixing the dataset $D_1$, for each $t \in [H]$ we bound the convergence rate of the empirical distribution estimate (computed using $D_2$) of $\Pr_{\widehat{\pi}}[\mathcal{E}_{D_1}, s_t = s, a_t = a]$ to the population in $\ell_1$ distance, and

(ii) we show that the resulting bound (which is a function of $D_1$) has small expectation and converges to 0 quickly.

This establishes the following bound on the expected suboptimality incurred by MIMIC-MD in Theorem 4.1 (a) and the policy in Theorem 4.2 which approximately optimizes OPT.

**Lemma A.12.**

$$\sum_{s\in\mathcal{S}} \sum_{a\in\mathcal{A}} \sum_{t=1}^{H} \mathbb{E}\left[ \left| \Pr_{\pi^*}\left[ \mathcal{T}_t^{\mathcal{E}_{D_1}}(s,a) \right] - \frac{\sum_{tr\in D_2} \mathbb{1}(tr \in \mathcal{T}_t^{\mathcal{E}_{D_1}}(s,a))}{|D_2|} \right| \right] \le \sqrt{\frac{8|\mathcal{S}|H^2}{N}} \wedge \frac{8}{3} \frac{|\mathcal{S}|H^{\frac{3}{2}}}{N}$$

(34)

To show the high probability guarantee on MIMIC-MD in Theorem 4.1 (b), the key approach is similar. However, we instead

(i) fix $D_1$ and use sub-Gaussian concentration [BLM13] to establish high probability deviation bounds on the empirical estimate of $\Pr_{\widehat{\pi}}[\mathcal{E}_{D_1}, s_t = s, a_t = a]$, and

(ii) use missing mass concentration (Theorem A.2) to show that the resulting deviations (which are a function of $D_1$) concentrate.

**Lemma A.13.** *Fix $\delta \in (0, \min\{1, H/5\})$. Then, with probability $\ge 1 - \delta$,*

$$\sum_{s\in\mathcal{S}} \sum_{a\in\mathcal{A}} \sum_{t=1}^{H} \left| \Pr_{\pi^*}\left[ \mathcal{T}_t^{D_1}(s,a) \right] - \frac{\sum_{tr\in D_2} \mathbb{1}(tr \in \mathcal{T}_t^{D_1}(s,a))}{|D_2|} \right|$$

$$\lesssim \frac{|\mathcal{S}|H^{3/2}}{N} \left( 1 + \frac{3\log(2|\mathcal{S}|H/\delta)}{\sqrt{|\mathcal{S}|}} \right)^{1/2} \sqrt{\log(2|\mathcal{S}|H/\delta)}.$$

(35)

The high probability guarantee of MIMIC-MD follows suit by invoking Lemma A.11 and Lemma A.13.

## A.4 Proof of lower bounds

In this section we discuss the proof of Theorem 5.1 establishing lower bounds on the expected suboptimality in the no-interaction, active and known-transition settings.

### A.4.1 Active and no-interaction settings

In this section we discuss the proof of the lower bounds in Theorem 5.1 (a) for the no-interaction and active settings. We emphasize that the active setting is strictly a generalization of the no-interaction

setting: they are no different if the learner queries and plays the expert's action at each time while interacting with the MDP.

Formally, in the active setting, we assume the learner sequentially rolls out policies $\pi_1, \cdots \pi_N$ to generate trajectories $\mathsf{tr}_1, \cdots, \mathsf{tr}_N$. The learner is aware of the expert's action at each state visited in each trajectory $\mathsf{tr}_n$, however may or may not choose to play this action while rolling out $\pi_n$. We assume that the policy $\pi_n$ is learnt causally, and can depend on all the previous information collected by the learner: the trajectories $\mathsf{tr}_1, \cdots, \mathsf{tr}_{n-1}$, as well as the expert's policy at each state visited in these trajectories.

**Notation:** We use $D = \mathsf{tr}_1, \cdots, \mathsf{tr}_n$ to denote the trajectories collected by the learner by rolling out $\pi_1, \cdots, \pi_N$. In addition the learner exactly knows the expert's policy $\pi_t^*(\cdot|s)$ at all states $s \in \mathcal{S}_t(D)$. We also define $A = \{\pi_t^*(\cdot|s) : t \in [H], s \in \mathcal{S}_t(D)\}$ as the expert's policy at states visited in $D$, which is also known to the learner by virtue of actively querying the expert.

The expert policy is deterministic in the lower bound instances we construct. Therefore, we define $\Pi_{\mathrm{mimic}}(D, A)$ (similar to $\Pi_{\mathrm{mimic}}(D)$ in eq. (4)) as the family of deterministic policies which mimics the expert on the states visited in $D$. Namely,

$$\Pi_{\mathrm{mimic}}(D, A) \triangleq \left\{\pi \in \Pi_{\det} : \forall t \in [H], s \in \mathcal{S}_t(D), \ \pi_t(s) = \pi_t^A(s)\right\} \tag{36}$$

where $\delta_{\pi_t^A(s)}$ is the policy observed by the learner upon actively querying the expert in a trajectory that visits $s$ at time $t$. Informally, $\Pi_{\mathrm{mimic}}(D, A)$ is the family of expert policies which are "compatible" with the dataset $(D, A)$ collected by the learner.

Define $\mathbb{M}_{\mathcal{S}, \mathcal{A}, H}$ as the family of MDPs over state space $\mathcal{S}$, action space $\mathcal{A}$ and with episode length $H$.

In order to prove the lower bound on the expected suboptimality of any learner $\widehat{\pi}(D, A)$, it suffices lower bound the Bayes expected suboptimality. Namely, it suffices to find a joint distribution $\mathcal{P}$ over MDPs and expert policies supported on $\mathbb{M}_{\mathcal{S}, \mathcal{A}, H} \times \Pi_{\det-\exp}$ such that,

$$\mathbb{E}_{(\pi^*, \mathcal{M}) \sim \mathcal{P}}\left[J_{\mathcal{M}}(\pi^*) - \mathbb{E}\left[J_{\mathcal{M}}(\widehat{\pi}(D, A))\right]\right] \gtrsim \min\left\{H, \frac{|\mathcal{S}|H^2}{N}\right\}. \tag{37}$$

**Construction of $\mathcal{P}$:** First we choose the expert's policy uniformly from $\Pi_{\det}$. That is, for each $t \in [H]$ and $s \in \mathcal{S}$, $\pi_t^*(s) \sim \mathrm{Unif}(\mathcal{A})$. Conditioned on $\pi^*$, the distribution over MDPs induced by $\mathcal{P}$ is deterministic and given by the MDP $\mathcal{M}[\pi^*]$ in fig. 2. The $\mathcal{M}[\pi^*]$ is defined with respect to a fixed initial distribution over states $\rho = \{\xi, \cdots, \xi, 1-(|\mathcal{S}|-2)\xi, 0\}$ where $\xi = \frac{1}{N+1}$. In addition, there is a special state $b \in \mathcal{S}$ which we refer to as the "bad state". At each state $s \in \mathcal{S} \setminus \{b\}$, choosing the expert's action renews the state in the initial distribution $\rho$ and dispenses a reward of $1$, while any other choice of action deterministically transitions to the bad state and offers no reward. In addition, the bad state is absorbing and dispenses no reward irrespective of the choice of action. That is,

$$P_t(\cdot|s, a) = \begin{cases} \rho, & s \in \mathcal{S} \setminus \{b\}, \ a = \pi_t^*(s) \\ \delta_b, & \text{otherwise,} \end{cases} \tag{38}$$

and the reward function of the MDP is given by,

$$\mathbf{r}_t(s, a) = \begin{cases} 1, & s \in \mathcal{S} \setminus \{b\}, \ a = \pi_t^*(s) \\ 0, & \text{otherwise.} \end{cases} \tag{39}$$

We first state a simple consequence of the construction of the MDP instances and $\mathcal{P}$.

**Lemma A.14.** *Consider any policy $\pi^* \in \Pi_{\det}$. Then, the value of $\pi^*$ on the MDP $\mathcal{M}[\pi^*]$ is $H$.*

*Proof.* Playing the expert's action at any state in $\mathcal{S} \setminus \{b\}$ is the only way to accrue non-zero reward, and in fact accrues a reward of $1$. In addition, note that the expert never visits the bad state $b$ by virtue of the distribution $\rho$ placing no mass on $b$. Therefore, the value of $\pi^*$ on the MDP $\mathcal{M}[\pi^*]$ is $H$. $\quad\square$

The intuition behind the lower bound construction is as follows. Although the learner can actively query the expert, at the states unvisited in the dataset $D$, the learner has no idea about the expert's policy or the transitions induced under different actions. Intuitively it is clear that the learner cannot

Figure 2: MDP template when $N_{\text{sim}} = 0$: Upon playing the expert's (green) action at any state except $b$, learner is renewed in the initial distribution $\rho = \{\xi, \cdots, \xi, 1-(|\mathcal{S}|-2)\xi, 0\}$ where $\xi = \frac{1}{N+1}$. Any other choice of action (red) deterministically transitions the state to $b$.

guess the expert's action with probability $\geq 1/2$ at such states, a statement which we prove by leveraging the Bayesian construction. In turn, the learner is forced to visit the bad state $b$ at the next point in the episode, and then on collects no reward.

Therefore, to bound the expected reward collected by a learner, it suffices to bound the probability that a learner visits a state unvisited in the expert dataset. The remainder of the proof is in showing that in this MDP construction, in expectation any learner visits such states with probability $\epsilon \gtrsim |\mathcal{S}|/N$ at each point in an episode. Moreover, conditioned on the dataset $D$, these events occur independently across time. Thus informally, the expected suboptimality of a learner is lower bounded by,

$$H\epsilon + (H-1)\epsilon(1-\epsilon) + \cdots + (1-\epsilon)^H \gtrsim \min\{H, H^2\epsilon\}. \tag{40}$$

where $\epsilon = |\mathcal{S}|/N$.

We return to a more formal exposition of the proof of the lower bound. Recall that our objective is to lower bound the Bayes expected suboptimality of $\widehat{\pi}$. Invoking Lemma A.14, the objective is to lower bound

$$\mathbb{E}_{(\pi^*, \mathcal{M}) \sim \mathcal{P}} \Big[ H - \mathbb{E}\Big[ J_{\mathcal{M}}(\widehat{\pi}(D, A)) \Big] \Big] \tag{41}$$

To this end, we first try to understand the conditional distribution of the expert's policy given the dataset $(D, A)$ collected by the learner. Recall that the dataset $D$ contains trajectories generated by rolling out a sequence of policies $\pi_1, \cdots, \pi_n$, and $A$ captures the expert's policy at states visited in $D$.

**Lemma A.15.** *Conditioned on the dataset $(D, A)$ collected by the learner, the expert's deterministic policy $\pi^*$ is distributed $\sim \text{Unif}(\Pi_{\text{mimic}}(D, A))$. In other words, at each state visited in the expert dataset, the expert's choice of action is fixed as the one returned when the expert was actively queried at this state. At the remaining states, the expert's choice of action is sampled uniformly from $\mathcal{A}$.*

**Definition A.3.** *Define $\mathcal{P}(D, A)$ as the joint distribution of $(\pi^*, \mathcal{M})$ conditioned on the dataset $(D, A)$ collected by the learner. In particular, $\pi^* \sim \text{Unif}(\Pi_{\text{mimic}}(D, A))$ and $\mathcal{M} = \mathcal{M}[\pi^*]$.*

From Lemma A.15 and the definition of $\mathcal{P}(D, A)$ in Definition A.3, we apply Fubini's theorem to give,

$$\mathbb{E}_{(\pi^*, \mathcal{M}) \sim \mathcal{P}} \Big[ H - \mathbb{E}\left[ J_{\mathcal{M}}(\widehat{\pi}) \right] \Big] = \mathbb{E} \Big[ \mathbb{E}_{(\pi^*, \mathcal{M}) \sim \mathcal{P}(D, A)} \left[ H - J_{\mathcal{M}}(\widehat{\pi}(D, A)) \right] \Big] \tag{42}$$

Next we relate this to the first time the learner visits a state unobserved in $D$.

**Lemma A.16.** *Define the stopping time $\tau$ as the first time $t$ that the learner encounters a state $s_t \neq b$ that has not been visited in $D$ at time $t$. That is,*

$$\tau = \begin{cases} \inf\{t : s_t \notin \mathcal{S}_t(D) \cup \{b\}\} & \exists t : s_t \notin \mathcal{S}_t(D) \cup \{b\} \\ H & \textit{otherwise}. \end{cases} \tag{43}$$

*Then, conditioned on the dataset $(D, A)$ collected by the learner,*

$$\mathbb{E}_{(\pi^*, \mathcal{M}) \sim \mathcal{P}(D, A)} \Big[ J(\pi^*) - \mathbb{E}\left[ J(\widehat{\pi}) \right] \Big] \geq \left(1 - \frac{1}{|\mathcal{A}|}\right) \mathbb{E}_{(\pi^*, \mathcal{M}) \sim \mathcal{P}(D, A)} \Big[ \mathbb{E}_{\widehat{\pi}(D, A)} \left[ H - \tau \right] \Big] \tag{44}$$

Figure 3: MDP template when $N_{\text{sim}} \to \infty$, Each state is absorbing, initial distribution is given by $\{\xi, \cdots, \xi, 1 - (|\mathcal{S}| - 1)\xi\}$ where $\xi = \frac{1}{N+1}$

Plugging the result of Lemma A.16 into eq. (42), we have that,

$$\mathbb{E}_{(\pi^*, \mathcal{M}) \sim \mathcal{P}} \left[ J(\pi^*) - \mathbb{E}\left[ J(\widehat{\pi}) \right] \right] \geq \left( 1 - \frac{1}{|\mathcal{A}|} \right) \mathbb{E} \left[ \mathbb{E}_{(\pi^*, \mathcal{M}) \sim \mathcal{P}(D,A)} \left[ \mathbb{E}_{\widehat{\pi}} \left[ H - \tau \right] \right] \right] \tag{45}$$

$$\overset{(i)}{\geq} \left( 1 - \frac{1}{|\mathcal{A}|} \right) \frac{H}{2} \mathbb{E} \left[ \mathbb{E}_{(\pi^*, \mathcal{M}) \sim \mathcal{P}(D,A)} \left[ \Pr_{\widehat{\pi}} \left[ \tau \leq \lfloor H/2 \rfloor \right] \right] \right] \tag{46}$$

$$= \left( 1 - \frac{1}{|\mathcal{A}|} \right) \frac{H}{2} \mathbb{E}_{(\pi^*, \mathcal{M}) \sim \mathcal{P}} \left[ \mathbb{E} \left[ \Pr_{\widehat{\pi}} \left[ \tau \leq \lfloor H/2 \rfloor \right] \right] \right] \tag{47}$$

where $(i)$ uses Markov's inequality and the last equation uses Fubini's theorem.

The last remaining element of he proof is to indeed bound the probability that the learner visits a state unobserved in the dataset before time $\lfloor H/2 \rfloor$. In Lemma A.17 we prove that for any learner $\widehat{\pi}$, $\mathbb{E}_{(\pi^*, \mathcal{M}) \sim \mathcal{P}} \left[ \mathbb{E} \left[ \Pr_{\widehat{\pi}} \left[ \tau \leq \lfloor H/2 \rfloor \right] \right] \right]$ is lower bounded by $\gtrsim \min\{1, |\mathcal{S}| H / N\}$. Therefore,

$$\mathbb{E}_{(\pi^*, \mathcal{M}) \sim \mathcal{P}} \left[ J(\pi^*) - \mathbb{E}\left[ J(\widehat{\pi}) \right] \right] \gtrsim \left( 1 - \frac{1}{|\mathcal{A}|} \right) \frac{H}{2} \min \left\{ 1, \frac{|\mathcal{S}| H}{N} \right\}. \tag{48}$$

Since $\left( 1 - \frac{1}{|\mathcal{A}|} \right)$ is a constant for $|\mathcal{A}| \geq 2$ the statement of Theorem 5.1 follows.

**Lemma A.17.** *For any learner policy $\widehat{\pi}$,*

$$\mathbb{E}_{(\pi^*, \mathcal{M}) \sim \mathcal{P}} \left[ \mathbb{E} \left[ \Pr_{\widehat{\pi}} \left[ \tau \leq \lfloor H/2 \rfloor \right] \right] \right] \geq 1 - \left( 1 - \frac{|\mathcal{S}| - 2}{e(N+1)} \right)^{\lfloor H/2 \rfloor} \gtrsim \min \left\{ 1, \frac{|\mathcal{S}| H}{N} \right\}. \tag{49}$$

### A.4.2 Known-transition setting

As in the proof of Theorem 5.1 (a), in order to prove the lower bound on the expected suboptimality of any learner $\widehat{\pi}(D, A)$, it suffices lower bound the Bayes expected suboptimality. Namely, it suffices to find a joint distribution $\mathcal{P}$ over MDPs and expert policies supported on $\mathbb{M}_{\mathcal{S}, \mathcal{A}, H} \times \Pi_{\text{det}-\exp}$ such that,

$$\mathbb{E}_{(\pi^*, \mathcal{M}) \sim \mathcal{P}} \left[ J(\pi^*) - \mathbb{E}\left[ J(\widehat{\pi}(D, P)) \right] \right] \gtrsim \min \left\{ H, \frac{|\mathcal{S}| H}{N} \right\}. \tag{50}$$

**Construction of $\mathcal{P}$**  As in the proof of Item 5.1 (a), we first sample the expert's policy uniformly from $\Pi_{\text{det}}$. That is, for each $t \in [H]$ and $s \in \mathcal{S}$, the action $\pi_t^*(s)$ is drawn uniformly from $\mathcal{A}$. Conditioned on $\pi^*$, the distribution over MDPs induced by $\mathcal{P}$ is deterministic and given by the construction $\mathcal{M}[\pi^*]$ in fig. 2. $\mathcal{M}[\pi^*]$ is defined with initial distribution over states $\rho = \{\xi, \cdots, \xi, 1 - (|\mathcal{S}| - 1)\xi\}$ where $\xi = \frac{1}{N+1}$. Each state $s \in \mathcal{S}$ is absorbing in $\mathcal{M}[\pi^*]$. Formally, for each $s \in \mathcal{S}$ the transition function of $\mathcal{M}[\pi^*]$ is,

$$P_t(\cdot|s, a) = \delta_s. \tag{51}$$

At any state $s$, choosing the expert's action $\pi_t^*(s)$ returns a reward of $1$, while any other choice of action offers $0$ reward.

$$\mathbf{r}_t(s, a) = \begin{cases} 1, & a = \pi_t^*(s) \\ 0, & \text{otherwise.} \end{cases} \tag{52}$$

Note that all the MDPs $\mathcal{M}[\pi^*]$ for $\pi^* \in \Pi_{\text{det}}$ share a common set of transition functions and initial state distribution. Therefore, fixing $P$ and $\rho$, we define $\mathcal{P}'$ to be the joint distribution over expert

policies and reward functions induced by $\mathcal{P}$. Then the objective is to lower bound the Bayes expected suboptimality,

$$\mathbb{E}_{(\pi^*, \mathbf{r}) \sim \mathcal{P}'} \left[ J_{\mathbf{r}}(\pi^*) - \mathbb{E}\left[J_{\mathbf{r}}(\widehat{\pi}(D, P))\right] \right]. \tag{53}$$

In this construction, it is yet again the case that the expert's policy $\pi^*$ collects maximum reward $H$ on $\mathcal{M}[\pi^*]$.

**Lemma A.18.** *Consider any policy $\pi^* \in \Pi_{\text{det}}$. Then, the value of $\pi^*$ on the MDP $\mathcal{M}[\pi^*]$ is $H$.*

*Proof.* At each state visited $\pi^*$ plays the only action which accrues a reward of 1. By accumulating a local reward of 1 at each step, $\pi^*$ has value equal to $H$ on the MDP $\mathcal{M}[\pi^*]$. $\qquad \square$

With this explanation, invoking Lemma A.18 shows that out objective is to now lower bound,

$$\mathbb{E}_{(\pi^*, \mathbf{r}) \sim \mathcal{P}'} \left[ H - \mathbb{E}\left[J_{\mathbf{r}}(\widehat{\pi}(D, P))\right] \right]. \tag{54}$$

Similar to Lemma A.15, we can compute the conditional distribution of the expert's policy (which marginally follows the uniform prior) given the expert dataset $D$.

**Lemma A.19.** *Conditioned on $D$, the distribution of the expert policy $\pi^*$ is uniform over the family of deterministic policies $\Pi_{\text{mimic}}(D)$ (as defined in eq. (4)).*

For brevity of notation, we define this conditional distribution of the expert policy given the dataset $D$ by $\mathcal{P}'(D)$.

**Definition A.4.** *Define $\mathcal{P}'(D)$ as the joint distribution of $(\pi^*, \mathbf{r})$ conditioned on the expert dataset $D$. In particular, $\pi^* \sim \text{Unif}(\Pi_{\text{mimic}}(D))$ and $\mathbf{r} = \mathbf{r}[\pi^*]$.*

From Lemma A.19 and Definition A.4 and applying Fubini's theorem,

$$\mathbb{E}_{(\pi^*, \mathbf{r}) \sim \mathcal{P}'} \left[ \mathbb{E}\left[H - J_{\mathbf{r}}(\widehat{\pi}(D, P))\right] \right] = \mathbb{E}\left[ \mathbb{E}_{(\pi^*, \mathbf{r}) \sim \mathcal{P}'(D)}\left[H - J_{\mathbf{r}}(\widehat{\pi}(D, P))\right] \right] \tag{55}$$

Fixing the expert dataset $D$, we subsequently show that the suboptimality incurred by the learner is $\Omega(H)$ if initialized in a state unobserved in the expert dataset $D$. The key intuition is to identify that here the learner's knowledge of the transition function plays no role as each state in the MDP is absorbing. Therefore, once again at states unvisited in the expert dataset, the learner cannot guess the expert's action with high probability at states, leading to errors that growing linearly in $H$.

**Lemma A.20.** *For any learner's policy $\widehat{\pi}$ conditioned on the expert dataset $D$,*

$$\mathbb{E}_{(\pi^*, \mathbf{r}) \sim \mathcal{P}'(D)} \left[ H - J_{\mathbf{r}}(\widehat{\pi}(D, P)) \right] \geq H \left(1 - \frac{1}{|\mathcal{A}|}\right) \left(1 - \rho(\mathcal{S}_1(D))\right) \tag{56}$$

Therefore, from Lemma A.20 and eq. (55),

$$\mathbb{E}_{(\pi^*, \mathbf{r}) \sim \mathcal{P}'} \left[ \mathbb{E}\left[H - J_{\mathbf{r}}(\widehat{\pi}(D, P))\right] \right] \geq H \left(1 - \frac{1}{|\mathcal{A}|}\right) \mathbb{E}\left[1 - \rho(\mathcal{S}_1(D))\right], \tag{57}$$

The last ingredient left to show is that the probability mass on states unobserved in the expert dataset, $1 - \rho(\mathcal{S}_1(D))$, is not too small in expectation. Here we realize that this boils down to calculating the expected missing mass of the distribution $\rho$ given $N$ samples drawn independently. By construction of $\rho$, we show that this is $\gtrsim |\mathcal{S}|/N$ in expectation.

**Lemma A.21.** $\mathbb{E}[1 - \rho(\mathcal{S}_1(D))] \geq \frac{|\mathcal{S}|-1}{e(N+1)}$.

Plugging Lemma A.21 back into eq. (57) certifies a lower bound on the Bayes expected suboptimality of any learner $\widehat{\pi}$. This implies the existence of an MDP on which the learner's expected suboptimality is $\gtrsim |\mathcal{S}|H/N$.

## Appendix B

## Contents

### B.1 Missing proofs for analysis of behavior cloning

#### B.1.1 Proof of Lemma A.1

Since the expert dataset $D$ is composed of trajectories generated by i.i.d. rollouts of $\pi^*$, we have that $\Pr[s \notin \mathcal{S}_\tau(D)] = (1 - \Pr_{\pi^*}[s_\tau = s])^{|D|}$. Therefore,

$$\sum_{t=1}^{H} \sum_{s \in \mathcal{S}} \Pr_{\pi^*}[s_t = s] \Pr[s \notin \mathcal{S}_t(D)] \le \sum_{\tau=1}^{H} \sum_{s \in \mathcal{S}} \Pr_{\pi^*}[s_\tau = s] \left(1 - \Pr_{\pi^*}[s_\tau = s]\right)^{|D|} \quad (58)$$

Noting that $\max_{x \in [0,1]} x(1-x)^N = \frac{1}{N+1}\left(1 - \frac{1}{N+1}\right)^N \le \frac{4}{9N}$, from eq. (58),

$$\sum_{\tau=1}^{H} \sum_{s \in \mathcal{S}} \Pr_{\pi^*}[s_\tau = s] \left(1 - \Pr_{\pi^*}[s_\tau = s]\right)^{|D|} \le \sum_{\tau=1}^{H} \sum_{s \in \mathcal{S}} \frac{4}{9|D|} \le \frac{4}{9} \frac{|\mathcal{S}|H}{|D|} \quad (59)$$

#### B.1.2 Proof of Theorem A.2

To prove this theorem, we invoke a result of [MOHG03] on the concentration of missing mass.

**Theorem B.1** (Concentration of missing mass [MOHG03]). *Consider an arbitrary distribution $\nu$ on $\mathcal{X}$, and let $X^N \overset{i.i.d.}{\sim} \nu$ be a dataset of $N$ samples drawn i.i.d. from $\nu$. Let $\beta \ge 0$ and $\sigma \ge 0$ be*

*constants such that $\sum_{x\in\mathcal{X}}(\nu(x))^2 e^{-(N-\beta)\nu(x)} \leq \sigma^2$. For any $0 \leq \varepsilon \leq \beta\sigma^2$, we have the following,*

$$\Pr\left(\mathfrak{m}_0(\nu, X^N) - \mathbb{E}[\mathfrak{m}_0(\nu, X^N)] \geq \varepsilon\right) \leq \exp\left(-\frac{\varepsilon^2}{2\sigma^2}\right). \tag{60}$$

We prove Theorem A.2 by an appropriate choice of parameters $\beta, \sigma^2$ and $\epsilon$ (as functions of the confidence parameter $\delta$). In particular, choose $\beta = N - \frac{N}{\sqrt{\log(1/\delta)}} \geq \frac{N}{3}$. For this choice of $\beta$,

$$\sum_{x\in\mathcal{X}}(\nu(x))^2 e^{-(N-\beta)\nu(x)} = \sum_{x\in\mathcal{X}}(\nu(x))^2 e^{-\frac{N}{\sqrt{\log(1/\delta)}}\nu(x)}, \tag{61}$$

$$\leq |\mathcal{X}| \sup_{\nu\in[0,1]} \nu^2 e^{-\frac{N}{\sqrt{\log(1/\delta)}}\nu}, \tag{62}$$

$$\overset{(i)}{=} |\mathcal{X}|\left(4e^{-2}\frac{\log(1/\delta)}{N^2}\right). \tag{63}$$

where $(i)$ involves computing the supremum explicitly by differentiation. Therefore, for $\beta = N - \frac{N}{\sqrt{\log(H/\delta)}}$, a feasible choice of $\sigma^2$ in Theorem B.1 that upper bounds $\sum_{x\in\mathcal{X}}(\nu(x))^2 e^{-(N-\beta)\nu(x)}$ is $\frac{3|\mathcal{X}|\log(1/\delta)}{N^2}$. Choose $\varepsilon = \frac{3\sqrt{|\mathcal{X}|}\log(1/\delta)}{N}$ (note that this choice satisfies $\varepsilon \leq \beta\sigma^2$ since $\beta \geq N/3$ and $\sigma^2 = \frac{9|\mathcal{X}|\log(1/\delta)}{N^2}$). Invoking Theorem B.1 with this choice of $\beta, \sigma^2$ and $\epsilon$,

$$\Pr\left(\mathfrak{m}_0(\nu, X^N) - \mathbb{E}[\mathfrak{m}_0(\nu, X^N)] \geq \frac{3\sqrt{|\mathcal{X}|}\log(1/\delta)}{N}\right) \leq \exp\left(-\frac{\left(3\sqrt{|\mathcal{X}|}N^{-1}\log(1/\delta)\right)^2}{9|\mathcal{X}|N^{-2}\log(1/\delta)}\right) = \delta. \tag{64}$$

This proves Theorem A.2.

### B.1.3 Proof of Lemma A.3

We decompose $\sum_{\tau=1}^{H}\sum_{s\in\mathcal{S}}\Pr_{\pi^*}[s_\tau = s]\mathbb{1}(s \notin \mathcal{S}_\tau(D))$ as $\sum_\tau Z_\tau$ where $Z_\tau = \sum_{s\in\mathcal{S}}\Pr_{\pi^*}[s_\tau = s]\mathbb{1}(s \notin \mathcal{S}_\tau(D))$. Observe that for each fixed $\tau$, $Z_\tau$ is in fact the missing mass of the distribution over states at time $\tau$ rolling out $\pi^*$, given $N$ samples from the distribution. Applying the missing mass concentration inequality from Theorem A.2, with probability $\geq 1 - \delta/H$,

$$Z_\tau - \mathbb{E}[Z_\tau] \leq \frac{3\sqrt{|\mathcal{S}|}\log(H/\delta)}{N}. \tag{65}$$

Therefore, by union bounding, with probability $\geq 1 - \delta$,

$$\sum_{\tau=1}^{H} Z_\tau \leq \sum_{\tau=1}^{H}\mathbb{E}[Z_\tau] + H \cdot \frac{3\sqrt{|\mathcal{S}|}\log(H/\delta)}{N}. \tag{66}$$

Using $\sum_{\tau=1}^{H} Z_\tau = \sum_{\tau=1}^{H}\sum_{s\in\mathcal{S}}\Pr_{\pi^*}[s_\tau = s]\mathbb{1}(s \notin \mathcal{S}_\tau(D))$ and applying Lemma A.1 to claim that $\sum_{\tau=1}^{H}\mathbb{E}[Z_\tau] \leq 4|\mathcal{S}|H/9N$ completes the proof.

### B.2 Reduction of IL to supervised learning under TV distance (Lemma A.4)

For each $\tau \in [H]$, define the policy $\widetilde{\pi}^\tau = \{\pi_1^*, \cdots, \pi_\tau^*, \widehat{\pi}_{\tau+1}, \cdots, \widehat{\pi}_H\}$ with $\widetilde{\pi}^0 = \widehat{\pi}$. $\widetilde{\pi}^\tau$ plays the expert's policy till time $\tau$ and the learner's policy for the remainder of the episode. Then,

$$J(\pi^*) - J(\widehat{\pi}) = \sum_{\tau=1}^{H} J(\widetilde{\pi}^\tau) - J(\widetilde{\pi}^{\tau-1}) \tag{67}$$

For any fixed $\tau \in [H]$, observe that $\widetilde{\pi}^\tau$ and $\widetilde{\pi}^{\tau-1}$ roll out the same policy till time $\tau - 1$. Therefore the expected reward collected until time $\tau - 1$ for both policies is the same. By linearity of expectation,

$$J(\widetilde{\pi}^\tau) - J(\widetilde{\pi}^{\tau-1}) = \sum_{t=\tau}^{H}\mathbb{E}_{\widetilde{\pi}^\tau}\left[\mathbf{r}_t(s_t, a_t)\right] - \mathbb{E}_{\widetilde{\pi}^{\tau-1}}\left[\mathbf{r}_t(s_t, a_t)\right]. \tag{68}$$

Now fix some $t \geq \tau$ and consider $\mathbb{E}_{\widetilde{\pi}^\tau}\left[\mathbf{r}_t(s_t, a_t)\right] - \mathbb{E}_{\widetilde{\pi}^{\tau-1}}\left[\mathbf{r}_t(s_t, a_t)\right]$. First observe that,

$$\mathbb{E}_{\widetilde{\pi}^{\tau-1}}\left[\mathbf{r}_t(s_t, a_t)\right] = \mathbb{E}_{\substack{s_\tau \sim f_{\pi^*}^\tau \\ a_\tau \sim \widehat{\pi}_\tau(\cdot|s_\tau)}}\left[\mathbb{E}_{\widetilde{\pi}^{\tau-1}}\left[\mathbf{r}_t(s_t, a_t)|s_\tau, a_\tau\right]\right] \tag{69}$$

$$= \sum_{s \in \mathcal{S}} \sum_{a \in \mathcal{A}} f_{\pi^*}^\tau(s)\, \widehat{\pi}_\tau(a|s)\, \mathbb{E}_{\widetilde{\pi}^{\tau-1}}\left[\mathbf{r}_t(s_t, a_t)|s_\tau = s, a_\tau = a\right] \tag{70}$$

$$= \sum_{s \in \mathcal{S}} \sum_{a \in \mathcal{A}} f_{\pi^*}^\tau(s)\, \widehat{\pi}_\tau(a|s)\, \mathbb{E}_{\widehat{\pi}}\left[\mathbf{r}_t(s_t, a_t)|s_\tau = s, a_\tau = a\right] \tag{71}$$

where in the last equation we use the fact that $\widetilde{\pi}^{\tau-1}$ rolls out $\widehat{\pi}$ time $\tau$ onwards, and the fact that we condition on the state visited and action played at time $\tau$. Moreover, we also use the fact that $\mathbf{r}_t(s_t, a_t)$ only depends on $(s_t, a_t)$ which appears at time $t \geq \tau$. Noting that $\widetilde{\pi}^\tau = (\pi_1^*, \cdots, \pi_\tau^*, \widehat{\pi}_{\tau+1}, \cdots, \widehat{\pi}_H)$, a similar decomposition gives,

$$\mathbb{E}_{\widetilde{\pi}^\tau}\left[\mathbf{r}_t(s_t, a_t)\right] = \sum_{s \in \mathcal{S}} \sum_{a \in \mathcal{A}} f_{\pi^*}^\tau(s)\, \pi_\tau^*(a|s)\, \mathbb{E}_{\widetilde{\pi}^\tau}\left[\mathbf{r}_t(s_t, a_t)|s_\tau = s, a_\tau = a\right] \tag{72}$$

$$= \sum_{s \in \mathcal{S}} \sum_{a \in \mathcal{A}} f_{\pi^*}^\tau(s)\, \pi_\tau^*(a|s)\, \mathbb{E}_{\widehat{\pi}}\left[\mathbf{r}_t(s_t, a_t)|s_\tau = s, a_\tau = a\right] \tag{73}$$

where in the last equation we similarly use the fact that $\widetilde{\pi}^\tau$ rolls out $\widehat{\pi}$ time $\tau + 1$ onwards, and the fact that we condition on the action played at time $\tau$. Subtracting eq. (71) from eq. (73),

$$\mathbb{E}_{\widetilde{\pi}^\tau}\left[\mathbf{r}_t(s_t, a_t)\right] - \mathbb{E}_{\widetilde{\pi}^{\tau-1}}\left[\mathbf{r}_t(s_t, a_t)\right]$$
$$\leq \sum_{s \in \mathcal{S}} f_{\pi^*}^\tau(s) \sum_{a \in \mathcal{A}} \mathbb{E}_{\widehat{\pi}}\left[\mathbf{r}_t(s_t, a_t)|s_\tau = s, a_\tau = a\right]\left(\pi_\tau^*(a|s) - \widehat{\pi}_\tau(a|s)\right) \tag{74}$$

Observe that $\mathbb{E}_{\widehat{\pi}}\left[\mathbf{r}_t(s_t, a_t)|s_\tau = s, a_\tau = a\right]$ is a function of $(s, a)$ and is bounded in $[0, 1]$ (since pointwise $0 \leq \mathbf{r}_t \leq 1$). Therefore,

$$\mathbb{E}_{\widetilde{\pi}^\tau}\left[\mathbf{r}_t(s_t, a_t)\right] - \mathbb{E}_{\widetilde{\pi}^{\tau-1}}\left[\mathbf{r}_t(s_t, a_t)\right] \leq \sum_{s \in \mathcal{S}} f_{\pi^*}^\tau(s) \sup_{g:\mathcal{A}\to[0,1]} \sum_{a \in \mathcal{A}} g(a)\left(\pi_\tau^*(a|s) - \widehat{\pi}_\tau(a|s)\right) \tag{75}$$

$$\overset{(i)}{=} \sum_{s \in \mathcal{S}} f_{\pi^*}^\tau(s)\mathsf{TV}\left(\pi_\tau^*(a|s), \widehat{\pi}_\tau(a|s)\right) \tag{76}$$

$$= \mathbb{E}_{s \sim f_{\pi^*}^\tau}\left[\mathsf{TV}\left(\pi_\tau^*(a|s), \widehat{\pi}_\tau(a|s)\right)\right]. \tag{77}$$

where $(i)$ uses the dual representation of TV distance. Summing over $t \geq \tau$ and $\tau \in [H]$ and invoking eqs. (67) and (68) we get,

$$J(\pi^*) - J(\widehat{\pi}) \leq H \sum_{\tau=1}^H \mathbb{E}_{s \sim f_{\pi^*}^\tau}\left[\mathsf{TV}\left(\pi_\tau^*(a|s), \widehat{\pi}_\tau(a|s)\right)\right]. \tag{78}$$

Using the definition of $\mathbb{T}_{\text{pop}}$ (eq. (19)) completes the proof.

## B.3 Missing proofs for Theorem 3.3

### B.3.1 Proof of Lemma A.5

Recall that we assume that the trajectories in the expert dataset are ordered arbitrarily as $\{\text{tr}_1, \cdots, \text{tr}_N\}$ where $\text{tr}_n = \{(s_1^n, a_1^n), \cdots, (s_H^n, a_H^n)\}$. $N_{t,s} = \{n \in [N] : s_t^n = s\}$ as defined in eq. (21) is the set of indices of trajectories in $D$ that visit the state $s$ at time $t$. In order to prove this result, suppose the learner's policy $\widehat{\pi}$

With this, we define the randomized policy $X^{\text{unif}}(D)$ as,

$$X_t^{\text{unif}}(\cdot|s) = \begin{cases} \delta_{a_t^{n(t,s)}} & \text{if } |N_{t,s}| \geq 1, \\ \text{Unif}(\mathcal{A}) & \text{otherwise.} \end{cases} \tag{79}$$

where each $n_{t,s}$ is a random variable independently sampled from $\text{Unif}(N_{t,s})$ whenever $N_{t,s} \neq \emptyset$. Note that fixing $D$ and $n(t, s)$ for all $t, s$ such that $N_{t,s} \neq \emptyset$, the random variable $X^{\text{unif}}$ is a fixed stochastic policy.

The policy $X^{\mathrm{unif}}(D)$ in a sense corresponds to just extracting the randomness in the actions chosen at visited states in the policy $\widehat{\pi}(D)$ returned by MIMIC-EMP. In particular, it is a short proof to see that the random variables $J(X^{\mathrm{unif}}(D))$ and $J(\widehat{\pi}(D))$ have the same expectation.

**Lemma B.2.** $\mathbb{E}[J(\widehat{\pi}(D))] = \mathbb{E}[J(X^{\mathrm{unif}}(D))]$.

*Proof.* Consider some trajectory $\mathrm{tr} = \{(s_1, a_1), \cdots, (s_H, a_H)\}$. Fixing the expert dataset $D$,

$$\mathbb{E}\left[\Pr_{X^{\mathrm{unif}}(D)}[\mathrm{tr}]\Big| D\right] = \mathbb{E}\left[\rho(s_1)\left(\prod_{t=1}^{H-1} X_t^{\mathrm{unif}}(a_t|s_t) P_t(s_{t+1}|s_t, a_t)\right) X_t^{\mathrm{unif}}(a_H|s_H)\right] \quad (80)$$

From eq. (79) and Algorithm 1, observe that $X_t^{\mathrm{unif}(\cdot|s)} = \widehat{\pi}(\cdot|s) = \mathrm{Unif}(\mathcal{A})$ at states $s : N_{t,s} = \emptyset$ (i.e. which were not visited in the expert dataset). Moreover, on the remaining states $X_t^{\mathrm{unif}}(a_t|s_t)$ is independently sampled from the empirical distribution over states at time $t$. In particular, this means that $\mathbb{E}[X_t^{\mathrm{unif}}(a_t|s_t)] = \widehat{\pi}_t(a_t|s_t)$. Plugging this in gives,

$$\mathbb{E}[\Pr_{X^{\mathrm{unif}}(D)}[\mathrm{tr}]] = \Pr_{\widehat{\pi}(D)}[\mathrm{tr}] \quad (81)$$

Multiplying both sides by $\sum_{t=1}^{H} \mathbf{r}_t(s_t, a_t)$, summing over all trajectories $\mathrm{tr}$ and taking expectation with respect to the expert dataset $D$ completes the proof. $\qquad\square$

First we provide an auxiliary result that is critical to showing that the policies $J(X^{\mathrm{unif}}(D))$ and $\pi^{\mathrm{first}}(D)$ have the same value in expectation.

To this end, first define $D_{\leq\tau,<\tau} = \{((s_1^n, a_1^n), \cdots, (s_{\tau-1}^n, a_{\tau-1}^n), s_\tau^n) : n \in [N]\}$ to be the truncation of the expert dataset $D$ till time $\tau$, excluding the actions played at this time. $D_{\leq\tau,\leq\tau}$ and other similar notations are defined analogously.

**Lemma B.3.** *Condition on $D_{\leq\tau,<\tau}$ which represents the truncation of trajectories in the expert dataset $D$ till the state visited at time $\tau$. At any state $s$ that is visited at least once in $D$ at time $\tau$ (namely with $|N_{\tau,s}| > 0$), the actions $\{a_\tau^n : n \in N_{\tau,s}\}$ played at trajectories that visit the state $s$ at time $\tau$ are drawn independently and identically $\sim \pi_\tau^*(\cdot|s)$.*

*Proof.* Recall that we condition on $D_{\leq\tau,<\tau}$ which captures trajectories in the expert dataset truncated till the state visited at time $\tau$. Since each trajectory $\mathrm{tr}_n \in [N]$ is rolled out independently, the action $a_\tau^n$ in each trajectory $\mathrm{tr}_n$ is drawn independently from $\pi_\tau^*(\cdot|s_\tau^n)$.

More importantly, conditioned on $D_{\leq\tau,<\tau}$ the states $s_\tau^n$ visited in different trajectories is determined. This implies that $N_{\tau,s}$ for $s \in \mathcal{S}$ is a measurable function of $D_{\leq\tau,<\tau}$.

These two statements together imply that states $s \in \mathcal{S}$ having $N_{\tau,s} > 0$ (which is a measurable function of $D_{\leq\tau,<\tau}$) are such that all the actions $\{a_\tau^n : n \in N_{\tau,s}\}$ are independent. $\qquad\square$

*Proof of Lemma A.5.* In order to prove this result, we use an inductive argument. The induction hypothesis is that the expected value of $X^{\mathrm{unif}}(D)$ and $\pi^{\mathrm{first}}(D)$ are the same, conditioned on the expert dataset till time $t$ and the actions from the empirical distribution sampled by $X^{\mathrm{unif}}(D)$ at different states till time $t$. We formalize this hypothesis in equations after first proving the base case. To recognize the fact that we prove the statement starting from $t = H$, we define $\mathcal{H}_H$ as the base case, and inductively prove $\mathcal{H}_{t-1}$ assuming the hypothesis $\mathcal{H}_t$.

First observe that,

$$\mathbb{E}\left[J(X^{\mathrm{unif}}(D))\Big|D_{\leq H,<H}, \Big\{n(t,s) \,\Big|\, t \leq H, \, s : N_{t,s} > 0\Big\}\right] = \mathbb{E}\left[J(\pi^{\mathrm{first}}(D))\Big|D_{\leq H,<H}\right] \quad (82)$$

This is because conditioned on $D_{<H,<H}$, the only randomness is in the actions that are played in the different trajectories at time $H$. By Lemma B.3 these are distributed i.i.d. $\sim \pi_t^*(\cdot|s)$. Taking expectation with respect to $\{n_{H,s} : t \leq H, \, s : N_{t,s} > 0\}$, results in proof of the base case for $t = H$,

$$\mathcal{H}_H : \mathbb{E}\left[J(X^{\mathrm{unif}}(D))\Big|D_{\leq H,<H}, \Big\{n(t,s) \,\Big|\, t < H, \, s : N_{t,s} > 0\Big\}\right] = \mathbb{E}\left[J(\pi^{\mathrm{first}}(D))\Big|D_{\leq H,<H}\right]$$

In general consider the hypothesis $\mathcal{H}_\tau$,

$$\mathcal{H}_\tau : \mathbb{E}\left[J(X^{\mathrm{unif}}(D))\Big|D_{\leq\tau,<\tau}, \Big\{n(t,s) \,\Big|\, t < \tau, \, s : N_{t,s} > 0\Big\}\right] = \mathbb{E}\left[J(\pi^{\mathrm{first}}(D))\Big|D_{\leq\tau,<\tau}\right].$$

Taking expectation with respect to $\{s_\tau^n : n \in [N]\}$, where conditionally $s_\tau^n \sim P_\tau(\cdot|s_{\tau-1}^n, a_{\tau-1}^n)$,

$$\mathbb{E}\left[J(X^{\text{unif}}(D))\Big| D_{<\tau,<\tau}, \left\{n(t,s) \,\Big|\, t < \tau, \, s : N_{t,s} > 0\right\}\right] = \mathbb{E}\left[J(\pi^{\text{first}}(D))\Big| D_{<\tau,<\tau}\right]. \quad (83)$$

Furthermore, taking expectation with respect to the actions $\{a_\tau^n : n \in [N]\}$ results in,

$$\mathbb{E}\left[J(X^{\text{unif}}(D))\Big| D_{<\tau,<\tau-1}, \left\{n(t,s) \,\Big|\, t < \tau, \, s : N_{t,s} > 0\right\}\right] = \mathbb{E}\left[J(\pi^{\text{first}}(D))\Big| D_{<\tau,<\tau-1}\right] \quad (84)$$

Invoking Lemma B.3, we see that the expectation on the LHS is the same irrespective of the choice of indices $\{n(\tau,s) : s : N_{\tau,s} > 0\}$ which the policy $X^{\text{unif}}$ chooses to play. Therefore,

$$\mathbb{E}\left[J(X^{\text{unif}}(D))\Big| D_{<\tau,<\tau-1}, \left\{n(t,s) \,\Big|\, t < \tau - 1, \, s : N_{t,s} > 0\right\}\right] = \mathbb{E}\left[J(\pi^{\text{first}}(D))\Big| D_{<\tau,<\tau-1}\right] \quad (85)$$

This proves the induction hypothesis $\mathcal{H}_{\tau-1}$ and consequently the hypothesis $\mathcal{H}_1$. Taking expectation on both sides of $\mathcal{H}_1$ with respect to $s_1^n \overset{\text{i.i.d.}}{\sim} \rho$ proves the claim. $\qquad\square$

### B.3.2 Proof of Lemma A.7

Fixing the table $\mathbf{T}^*$, the probability of observing the trajectory $\text{tr} = \{(s_1, a_1), \cdots, (s_H, a_H)\}$ under the deterministic policy $\pi^{\text{orc-first}}$ is,

$$\text{Pr}_{\pi^{\text{orc-first}}}(\text{tr}) = \rho(s_1)\left(\prod_{t=1}^{H-1} \mathbb{1}\left(a_t = \mathbf{T}^*_{t,s_t}(1)\right) P_t(s_{t+1}|s_t, a_t)\right) \mathbb{1}\left(a_H = \mathbf{T}^*_{H,s_H}(1)\right). \quad (86)$$

Since the actions $\mathbf{T}^*_{t,s_t}(1)$ are independently drawn from $\pi_t^*(\cdot|s_t)$, taking expectation, we see that

$$\mathbb{E}\left[\text{Pr}_{\pi^{\text{orc-first}}}(\text{tr})\right] = \rho(s_1)\left(\prod_{t=1}^{H-1} \pi_t^*(a_t|s_t) P_t(s_{t+1}|s_t, a_t)\right) \pi_H^*(a_H|s_H) = \text{Pr}_{\pi^*}(\text{tr}). \quad (87)$$

Multiplying both sides by $\sum_{t=1}^H \mathbf{r}_t(s_t, a_t)$ and summing over all trajectories completes the proof.

### B.3.3 Proof of Lemma A.9

Recall that the "failure" $\mathcal{E}$ is defined as the event that at some time $t \in [H]$, a state $s_t$ is visited such that $|N_{t,s_t}| = 0$, i.e. that was not visited in the expert dataset. By union bounding,

$$\mathbb{E}\left[\text{Pr}_{\pi^{\text{orc-first}}}[\mathcal{E}]\right] \le \sum_{t=1}^H \sum_{s \in \mathcal{S}} \mathbb{E}\left[\text{Pr}_{\pi^{\text{orc-first}}}[\mathcal{E}_{s,t}]\right], \quad (88)$$

where $\mathcal{E}_{s,t}$ is the event that a failure occurs at the state $s$ at time $t$, i.e. the state $s$ is visited at time $t$ and $|N_{t,s}| = 0$. $\mathcal{E}_{s,t}$ is the intersection of two events. Therefore we have the upper bound,

$$\mathbb{E}\left[\text{Pr}_{\pi^{\text{orc-first}}}[\mathcal{E}_{s,t}]\right] \le \min\left\{\mathbb{E}\left[\text{Pr}_{\pi^{\text{orc-first}}}[s_t = s]\right], \mathbb{E}\left[\text{Pr}_{\pi^{\text{orc-first}}}[|N_{s,t}| = 0]\right]\right\}. \quad (89)$$

Observe that these two terms in the minimum are easy to compute. Firstly, using eq. (87), we have that,

$$\mathbb{E}\left[\text{Pr}_{\pi^{\text{orc-first}}}[s_t = s]\right] = \text{Pr}_{\pi^*}[s_t = s]. \quad (90)$$

On the other hand,

$$\mathbb{E}\left[\text{Pr}_{\pi^{\text{orc-first}}}[|N_{s,t}| = 0]\right] = \mathbb{E}[\mathbb{1}(|N_{s,t}| = 0)] = (1 - \text{Pr}_{\pi^*}[s_t = s])^N \quad (91)$$

where the last equation uses Lemma A.6. Putting together eqs. (90) and (91) with eq. (89),

$$\mathbb{E}\left[\text{Pr}_{\pi^{\text{orc-first}}}[\mathcal{E}_{s,t}]\right] \le \min\left\{\text{Pr}_{\pi^*}[s_t = s], \left(1 - \text{Pr}_{\pi^*}[s_t = s]\right)^N\right\}. \quad (92)$$

In Lemma B.4 we show that the RHS is upper bounded by $\log(N)/N$. Therefore,

$$\mathbb{E}\left[\text{Pr}_{\pi^{\text{orc-first}}}[\mathcal{E}_{s,t}]\right] \le \frac{\log N}{N}. \quad (93)$$

Plugging back into eq. (88) completes the proof.

**Lemma B.4.** *For any $x \in [0,1]$ and $N > 1$, $\min\{x, (1-x)^N\} \leq \frac{\log N}{N}$.*

*Proof.* $x$ is an increasing function, while $(1-x)^N$ is decreasing. For $x = \frac{\log N}{N}$,

$$(1-x)^N = \left(1 - \frac{\log N}{N}\right)^N \leq e^{-\log N} \leq N^{-1} \tag{94}$$

Therefore for $x \geq \frac{\log(N)}{N}$, $\min\{x, (1-x)^N\} \leq \frac{1}{N}$. Therefore $\min\{x, (1-x)^N\} \leq \frac{\log N}{N}$. $\qquad\square$

## B.4 Missing proofs for Theorems 4.1 (a) and 4.1 (b)

### B.4.1 Proof of Lemma A.10

Observe that the complement $\mathcal{E}^c_{D_1}$ is the event that the policy under consideration only visits states that were visited in at least one trajectory in the expert dataset.

To prove the statement, it suffices to prove that fixing the expert dataset $D$,

$$\mathbb{E}_{\pi^*}\left[\mathbb{1}(\mathcal{E}^c_{D_1})\sum_{t=1}^{H}\mathbf{r}_t(s_t, a_t)\right] = \mathbb{E}_{\widehat{\pi}}\left[\mathbb{1}(\mathcal{E}^c_{D_1})\sum_{t=1}^{H}\mathbf{r}_t(s_t, a_t)\right]. \tag{95}$$

The learner mimics the expert at all the states observed in the expert dataset, i.e. having $|N_{t,s}| > 0$. Observe that when the event $\mathcal{E}^c_{D_1}$ occurs, all the states visited in a trajectory have $|N_{t,s}| > 0$. Thus, both expectations are carried out with respect to the same policy and are hence equal. More precisely, observe that,

$$\mathbb{E}_{\widehat{\pi}}\left[\mathbb{1}(\mathcal{E}^c_{D_1})\sum_{t=1}^{H}\mathbf{r}_t(s_t, a_t)\right] = \mathbb{E}_{\substack{s_1 \sim \rho,\ t \in [H], \\ s_{t+1} \sim P(\cdot|s_t, a_t)}}^{a_t \sim \widehat{\pi}_t(\cdot|s_t)}\left[\mathbb{1}(\mathcal{E}^c_{D_1})\sum_{t=1}^{H}\mathbf{r}_t(s_t, a_t)\right] \tag{96}$$

$$\overset{(i)}{=} \mathbb{E}_{\substack{s_1 \sim \rho,\ t \in [H], \\ s_{t+1} \sim P(\cdot|s_t, a_t)}}^{a_t \sim \pi^*_t(\cdot|s_t)}\left[\mathbb{1}(\mathcal{E}^c_{D_1})\sum_{t=1}^{H}\mathbf{r}_t(s_t, a_t)\right] \tag{97}$$

$$= \mathbb{E}_{\pi^*}\left[\mathbb{1}(\mathcal{E}^c_{D_1})\sum_{t=1}^{H}\mathbf{r}_t(s_t, a_t)\right] \tag{98}$$

where $(i)$ uses the fact that when $s_t \in \mathcal{S}_t(D_1)$ (as implied by $\mathcal{E}^c_{D_1}$), then, $\pi^*_t(\cdot|s_t) = \widehat{\pi}_t(\cdot|s_t)$.

### B.4.2 Proof of Lemma A.11

First observe that we can write the reward $\mathbf{r}_t(s_t, a_t)$ accrued in some trajectory at time $t$ equals $\sum_{s \in \mathcal{S}}\sum_{a \in \mathcal{A}}\mathbf{r}_t(s, a)\mathbb{1}((s_t, a_t) = (s, a))$. Therefore, from Lemma A.10,

$$J(\pi^*) - J(\widehat{\pi}) = \sum_{s \in \mathcal{S}}\sum_{a \in \mathcal{A}}\sum_{t=1}^{H}\mathbf{r}_t(s, a)\left(\Pr_{\pi^*}\left[\mathcal{E}_{D_1}, s_t=s, a_t=a\right] - \Pr_{\widehat{\pi}}\left[\mathcal{E}_{D_1}, s_t=s, a_t=a\right]\right)$$

$$\leq \sum_{s \in \mathcal{S}}\sum_{a \in \mathcal{A}}\sum_{t=1}^{H}\left|\Pr_{\pi^*}\left[\mathcal{E}_{D_1}, s_t=s, a_t=a\right] - \Pr_{\widehat{\pi}}\left[\mathcal{E}_{D_1}, s_t=s, a_t=a\right]\right|$$

$$= \sum_{s \in \mathcal{S}}\sum_{a \in \mathcal{A}}\sum_{t=1}^{H}\left|\Pr_{\pi^*}\left[\mathcal{T}_t^{\mathcal{E}_{D_1}}(s, a)\right] - \Pr_{\widehat{\pi}}\left[\mathcal{T}_t^{\mathcal{E}_{D_1}}(s, a)\right]\right| \tag{99}$$

where the inequality follows from the assumption that $0 \leq \mathbf{r}_t(s, a) \leq 1$ and the last equation follows from the definition $\mathcal{T}_t^{\mathcal{E}_{D_1}}(s, a) = \{\{(s_\tau, a_\tau)\}_{\tau=1}^{H}|s_t=s, a_t=a, \exists\tau\in[H]: s_\tau \notin \mathcal{S}_\tau(D_1)\}$ is the set of trajectories that visit $(s, a)$ at time $t$ and at some point $t'$ in the episode visit a state not visited in any trajectory at time $t'$ in $D_1$. Using the definition of the learner's policy $\widehat{\pi}$ in the optimization problem OPT and applying the triangle inequality,

$$J(\pi^*) - J(\widehat{\pi}) \leq \sum_{s \in \mathcal{S}}\sum_{a \in \mathcal{A}}\sum_{t=1}^{H}\left|\Pr_{\pi^*}\left[\mathcal{T}_t^{\mathcal{E}_{D_1}}(s, a)\right] - \frac{\sum_{\mathsf{tr}\in D_2}\mathbb{1}(\mathsf{tr} \in \mathcal{T}_t^{\mathcal{E}_{D_1}}(s, a))}{|D_2|}\right|$$

$$+ \left|\frac{\sum_{\mathsf{tr}\in D_2}\mathbb{1}(\mathsf{tr} \in \mathcal{T}_t^{\mathcal{E}_{D_1}}(s, a))}{|D_2|} - \Pr_{\widehat{\pi}}\left[\mathcal{T}_t^{\mathcal{E}_{D_1}}(s, a)\right]\right|. \tag{100}$$

Observe that the expert's policy $\pi^*$ is a feasible policy to the optimization problem OPT. Since $\widehat{\pi}$ is the minimizer of this optimization problem, we have the upper bound,

$$J(\pi^*) - J(\widehat{\pi}) \leq 2 \sum_{s \in \mathcal{S}} \sum_{a \in \mathcal{A}} \sum_{t=1}^{H} \left| \Pr_{\pi^*} \left[ \mathcal{T}_t^{\mathcal{E}_{D_1}}(s,a) \right] - \frac{\sum_{\mathsf{tr} \in D_2} \mathbb{1}(\mathsf{tr} \in \mathcal{T}_t^{\mathcal{E}_{D_1}}(s,a))}{|D_2|} \right|. \tag{101}$$

### B.4.3   Proof of Lemma A.12

Recall that the bound in Lemma A.11 applies when the expert dataset $D$ is fixed. Also note that we carry out sample splitting in Algorithm 2 to give datasets $D_1$ and $D_2$. Fixing the dataset $D_2$ taking expectation on both sides of Lemma A.11,

$$J(\pi^*) - \mathbb{E}\left[J(\widehat{\pi})\right] \leq 2 \sum_{s \in \mathcal{S}} \sum_{a \in \mathcal{A}} \sum_{t=1}^{H} \mathbb{E}\left[ \left| \Pr_{\pi^*} \left[ \mathcal{T}_t^{\mathcal{E}_{D_1}}(s,a) \right] - \frac{\sum_{\mathsf{tr} \in D_2} \mathbb{1}(\mathsf{tr} \in \mathcal{T}_t^{\mathcal{E}_{D_1}}(s,a))}{|D_2|} \right| \right].$$

By Jensen's inequality, we can upper bound by the quadratic deviation,

$$J(\pi^*) - \mathbb{E}\left[J(\widehat{\pi})\right] \leq$$

$$2 \sum_{s \in \mathcal{S}} \sum_{a \in \mathcal{A}} \sum_{t=1}^{H} \left( \mathbb{E}\left[ \left( \Pr_{\pi^*} \left[ \mathcal{T}_t^{\mathcal{E}_{D_1}}(s,a) \right] - \frac{\sum_{\mathsf{tr} \in D_2} \mathbb{1}(\mathsf{tr} \in \mathcal{T}_t^{\mathcal{E}_{D_1}}(s,a))}{|D_2|} \right)^2 \right] \right)^{1/2} \tag{102}$$

Observe that each trajectory $\mathsf{tr} \in D_2$ is generated by independently rolling out $\pi^*$. Therefore, $\frac{1}{|D_2|} \sum_{\mathsf{tr} \in D_2} \mathbb{1}(\mathsf{tr} \in \mathcal{T}_t^{\mathcal{E}_{D_1}}(s,a))$ is an unbiased estimate of $\Pr_{\pi^*}[\mathcal{T}_t^{\mathcal{E}_{D_1}}(s,a)]$. Therefore the variance in eq. (102) can be bounded as (here $\mathsf{tr}_1$ is an arbitrary trajectory in $D_2$),

$$J(\pi^*) - \mathbb{E}\left[J(\widehat{\pi})\right] \leq 2 \sum_{s \in \mathcal{S}} \sum_{a \in \mathcal{A}} \sum_{t=1}^{H} \left( \frac{1}{|D_2|} \mathrm{Var}\left[ \mathbb{1}(\mathsf{tr}_1 \in \mathcal{T}_t^{\mathcal{E}_{D_1}}(s,a)) \right] \right)^{1/2} \tag{103}$$

$$\leq 2 \sum_{s \in \mathcal{S}} \sum_{a \in \mathcal{A}} \sum_{t=1}^{H} \left( \frac{1}{|D_2|} \Pr_{\pi^*}\left[ \mathcal{T}_t^{\mathcal{E}_{D_1}}(s,a) \right] \right)^{1/2} \tag{104}$$

where the last inequality uses the fact that the variance of an indicator function is at most its mean, and that each $\mathsf{tr} \in D_2$ is independently drawn by rolling out $\pi^*$. Taking expectation with respect to the dataset $D_1$, and applying Jensen's inequality,

$$J(\pi^*) - \mathbb{E}\left[J(\widehat{\pi})\right] \leq 2 \sum_{s \in \mathcal{S}} \sum_{a \in \mathcal{A}} \sum_{t=1}^{H} \frac{1}{|D_2|^{1/2}} \left( \mathbb{E}\left[ \Pr_{\pi^*}\left[ \mathcal{T}_t^{\mathcal{E}_{D_1}}(s,a) \right] \right] \right)^{1/2} \tag{105}$$

$$= 2 \sum_{s \in \mathcal{S}} \sum_{t=1}^{H} \frac{1}{|D_2|^{1/2}} \left( \mathbb{E}\left[ \Pr_{\pi^*}\left[ \mathcal{E}_{D_1}, s_t = s, a_t = \pi_t^*(s_t) \right] \right] \right)^{1/2}, \tag{106}$$

where in the last equation, we use the fact that at each state $s$ the expert plays a fixed action $\pi_t^*(s)$ at time $t$. By an application of Cauchy Schwarz inequality,

$$J(\pi^*) - \mathbb{E}\left[J(\widehat{\pi})\right] \leq 2 \sum_{t=1}^{H} \frac{|\mathcal{S}|^{1/2}}{|D_2|^{1/2}} \left( \sum_{s \in \mathcal{S}} \mathbb{E}\left[ \Pr_{\pi^*}\left[ \mathcal{E}_{D_1}, s_t = s, a_t = \pi_t^*(s) \right] \right] \right)^{1/2} \tag{107}$$

$$\leq 2 \sum_{t=1}^{H} \frac{|\mathcal{S}|^{1/2}}{|D_2|^{1/2}} \left( \mathbb{E}\left[ \Pr_{\pi^*}\left[ \mathcal{E}_{D_1} \right] \right] \right)^{1/2}. \tag{108}$$

Therefore, to prove the result it suffices to bound $\mathbb{E}\left[\Pr_{\pi^*}\left[\mathcal{E}_{D_1}\right]\right]$, which we carry out in Lemma B.5. Here we show that it is upper bounded by $\lesssim 1 \wedge |\mathcal{S}|H/|D_1|$. Subsequently using $|D_1| = |D_2| = N/2$ completes the proof.

**Lemma B.5.** *The probability of failure under the expert's policy is upper bounded by,*

$$\mathbb{E}\left[ \Pr_{\pi^*}\left[ \mathcal{E}_{D_1} \right] \right] \leq \frac{4}{9} \frac{|\mathcal{S}|H}{|D_1|} \tag{109}$$

*Proof.* Conditioned on $D_1$, we decompose based on the first failure time (i.e. the first time the event $\mathcal{E}_{D_1}$ is satisfied),

$$\Pr_{\pi^*}\left[\mathcal{E}_{D_1}\Big|D_1\right] = \Pr_{\pi^*}\left[\exists t \in [H] : s_t \notin \mathcal{S}_t(D_1)\Big|D_1\right], \tag{110}$$

$$= \sum_{\tau=1}^{H} \Pr_{\pi^*}\left[\forall t < \tau,\ s_t \in \mathcal{S}_t(D_1), s_\tau \notin \mathcal{S}_\tau(D_1)\Big|D_1\right] \tag{111}$$

$$\leq \sum_{\tau=1}^{H} \Pr_{\pi^*}\left[s_\tau \notin \mathcal{S}_\tau(D_1)\Big|D_1\right] \tag{112}$$

$$= \sum_{\tau=1}^{H} \sum_{s \in \mathcal{S}} \Pr_{\pi^*}[s_\tau = s]\mathbb{1}(s \notin \mathcal{S}_\tau(D_1)) \tag{113}$$

Taking expectation with respect to the expert dataset,

$$\mathbb{E}\left[\Pr_{\pi^*}\left[\mathcal{E}_{D_1}\Big|D_1\right]\right] \leq \sum_{\tau=1}^{H} \sum_{s \in \mathcal{S}} \Pr_{\pi^*}[s_\tau = s]\Pr[s \notin \mathcal{S}_\tau(D_1)] \tag{114}$$

The proof of the claim immediately follows by invoking Lemma A.1. $\qquad\square$

### B.4.4   Proof of Lemma A.13

Starting from the bound in Lemma A.11 and using the fact that at each state $s$ the expert plays a fixed action $\pi_t^*(s)$ at time $t$,

$$J(\pi^*) - J(\widehat{\pi}) \leq 2\sum_{s \in \mathcal{S}} \sum_{t=1}^{H} \left|\Pr_{\pi^*}\left[\mathcal{T}_t^{\mathcal{E}_{D_1}}(s, \pi_t^*(s))\right] - \frac{\sum_{\text{tr} \in D_2} \mathbb{1}(\text{tr} \in \mathcal{T}_t^{\mathcal{E}_{D_1}}(s, \pi_t^*(s)))}{|D_2|}\right| \tag{115}$$

Observe that $\mathbb{1}(\text{tr} \in \mathcal{T}_t^{\mathcal{E}_{D_1}}(s, \pi_t^*(s)))$ is a sub-Gaussian random variable with variance bounded by its expectation. Therefore, by sub-Gaussian concentration [BLM13], for each $s \in \mathcal{S}$ and $t \in [H]$, conditioned on $D_1$, with probability $\geq 1 - \frac{\delta}{2|\mathcal{S}|H}$,

$$\left|\frac{\sum_{\text{tr} \in D_2} \mathbb{1}(\text{tr} \in \mathcal{T}_t^{\mathcal{E}_{D_1}}(s, \pi_t^*(s)))}{|D_2|} - \Pr_{\pi^*}\left[\mathcal{T}_t^{\mathcal{E}_{D_1}}(s, \pi_t^*(s))\right]\right|$$

$$\leq \left(\Pr_{\pi^*}\left[\mathcal{T}_t^{\mathcal{E}_{D_1}}(s, \pi_t^*(s))\right]\right)^{1/2}\sqrt{\frac{2\log(2|\mathcal{S}|H/\delta)}{|D_2|}} \tag{116}$$

By union bounding over $s \in \mathcal{S}$ and $t \in [H]$, conditioned on $D_1$ with probability $\geq 1 - \frac{\delta}{2}$,

$$\sum_{t=1}^{H} \sum_{s \in \mathcal{S}} \left|\frac{\sum_{\text{tr} \in D_2} \mathbb{1}(\text{tr} \in \mathcal{T}_t^{\mathcal{E}_{D_1}}(s, \pi_t^*(s)))}{|D_2|} - \Pr_{\pi^*}\left[\mathcal{T}_t^{\mathcal{E}_{D_1}}(s, \pi_t^*(s))\right]\right|$$

$$\leq \sum_{t=1}^{H} \left(\sum_{s \in \mathcal{S}} \Pr_{\pi^*}\left[\mathcal{T}_t^{\mathcal{E}_{D_1}}(s, \pi_t^*(s))\right]\right)^{1/2}\sqrt{\frac{2\log(2|\mathcal{S}|H/\delta)}{|D_2|}} \tag{117}$$

$$\leq H|\mathcal{S}|^{1/2}\left(\Pr_{\pi^*}\left[\mathcal{E}_{D_1}\right]\right)^{1/2}\sqrt{\frac{2\log(2|\mathcal{S}|H/\delta)}{|D_2|}} \tag{118}$$

Applying Lemma A.3, with probability $\geq 1 - \delta/2$,

$$\Pr_{\pi^*}[\mathcal{E}_{D_1}] \leq \frac{4|\mathcal{S}|H}{9|D_1|} + H\sqrt{\frac{\log(H/\delta)}{|D_1|}} \tag{119}$$

Therefore union bounding the events of eqs. (118) and (119), with probability $\geq 1 - \delta$,

$$\sum_{t=1}^{H} \sum_{s \in \mathcal{S}} \left| \frac{\sum_{\mathsf{tr} \in D_2} \mathbb{1}(\mathsf{tr} \in \mathcal{T}_t^{\mathcal{E}_{D_1}}(s, \pi_t^*(s)))}{|D_2|} - \Pr_{\pi^*} \left[ \mathcal{T}_t^{\mathcal{E}_{D_1}}(s, \pi_t^*(s)) \right] \right|$$

$$\leq H |\mathcal{S}|^{1/2} \left( \frac{4|\mathcal{S}|H}{9|D_1|} + H \sqrt{\frac{\log(H/\delta)}{|D_1|}} \right)^{1/2} \sqrt{\frac{2 \log(2|\mathcal{S}|H/\delta)}{|D_2|}} \tag{120}$$

$$\lesssim \frac{|\mathcal{S}|H^{3/2}}{N} \left( 1 + \sqrt{\frac{N \log(H/\delta)}{|\mathcal{S}|^2}} \right)^{1/2} \sqrt{\log(2|\mathcal{S}|H/\delta)} \tag{121}$$

using the assumption that $N \lesssim |\mathcal{S}|^2 / \log(H/\delta)$ in the last inequality completes the proof.

## B.5 Missing proofs for Theorem 5.1 (a)

### B.5.1 Proof of Lemma A.15

Fix some policy $\pi \in \Pi_{\mathrm{det}}$. Consider any time $t \in [H]$ and state $s \in \mathcal{S}_t(D)$ which is visited in some trajectory in the dataset at time $t$. If $\pi_t(s)$ does not match the action $\pi_t^A(s)$ revealed by actively querying the expert in a trajectory in $D$ that visits $s$ at time $t$, the likelihood of $\pi$ given $D$ is exactly $0$ (since the expert is deterministic). On the other hand, the conditional probability of observing $(D, A)$ does not depend on the expert's action on the states that were not observed in $D$, since no trajectory visits these states. Since on these states the expert's action marginally follows the uniform distribution over $\mathcal{A}$, the result immediately follows.

### B.5.2 Proof of Lemma A.16

In order to prove this result, define the auxiliary random time $\tau_b$ to be the first time the learner first encounters the state $b$ while rolling out a trajectory. If no such state is encountered, $\tau$ is defined as $H + 1$. Formally,

$$\tau_b = \begin{cases} \inf\{t : s_t = b\} & \exists t : s_t = b \\ H + 1 & \text{otherwise.} \end{cases}$$

Conditioning on the learner's dataset $(D, A)$, first observe that

$$H - \mathbb{E}_{(\pi^*, \mathcal{M}) \sim \mathcal{P}(D, A)} [J(\widehat{\pi})] = H - \mathbb{E}_{(\pi^*, \mathcal{M}) \sim \mathcal{P}(D, A)} \left[ \mathbb{E}_{\widehat{\pi}} \left[ \sum_{t=1}^{H} \mathbf{r}_t(s_t, a_t) \right] \right] \tag{122}$$

$$\geq \mathbb{E}_{(\pi^*, \mathcal{M}) \sim \mathcal{P}(D, A)} \left[ \mathbb{E}_{\widehat{\pi}} \left[ H - \tau_b + 1 \right] \right] \tag{123}$$

where the last inequality follows from the fact that $\mathbf{r}$ is bounded in $[0, 1]$, and the state $b$ is absorbing and offers $0$ reward irrespective of the choice of action. Fixing the dataset $(D, A)$ and the expert's policy $\pi^*$ (which determines the MDP $\mathcal{M}[\pi^*]$), we study $\mathbb{E}_{\widehat{\pi}(D,A)} [H - \tau_b + 1]$ and try to relate it to $\mathbb{E}_{\widehat{\pi}(D,A)} [H - \tau]$.

To this end, first observe that for any $t \leq H - 1$ and state $s \in \mathcal{S}$,

$$\Pr_{\widehat{\pi}} [\tau_b = t + 1, \tau = t, s_t = s] = \Pr_{\widehat{\pi}} [\tau_b = t + 1 | \tau = t, s_t = s] \Pr_{\widehat{\pi}} [\tau = t, s_t = s] \tag{124}$$

$$= \left( 1 - \widehat{\pi}_t(\pi_t^*(s)|s) \right) \Pr_{\widehat{\pi}} [\tau = t, s_t = s]. \tag{125}$$

where in the last equation, we use the fact that the learner must play an action other than $\pi_t^*(s_t)$ to visit $b$ at time $t + 1$. Next we take expectation with respect to the randomness of $\pi^*$ which conditioned on $(D, A)$ is drawn from $\mathrm{Unif}(\Pi_{\mathrm{mimic}}(D, A))$ which also specifies the underlying MDP $\mathcal{M}[\pi^*]$. Observe that the dependence of the second term $\Pr_{\widehat{\pi}} [\tau = t, s_t = s]$ on $\pi^*$ comes from the probability computed with the underlying MDP chosen as $\mathcal{M}[\pi^*]$. However observe that it only depends on the characteristics of $\mathcal{M}[\pi^*]$ till time $t - 1$ which are determined by $\pi_1^*, \cdots, \pi_{t-1}^*$. On the other hand, the first term $(1 - \widehat{\pi}_t(\pi_t^*(s)|s))$ depends only on $\pi_t^*$. As a consequence the two terms depend on a disjoint set of random variables, which are independent (since conditionally $\pi^* \sim \Pi_{\mathrm{mimic}}(D, A)$ defined in eq. (36))

Therefore taking expectation with respect to the randomness of $\pi^* \sim \text{Unif}(\Pi_{\text{mimic}}(D, A))$ and $\mathcal{M} = \mathcal{M}[\pi^*]$ (which defines the joint distribution $\mathcal{P}(D, A)$ in eq. (36)),

$$\mathbb{E}_{(\pi^*, \mathcal{M}) \sim \mathcal{P}(D,A)}\Big[\Pr_{\widehat{\pi}(D,A)}\big[\tau_b = t + 1, \tau = t, s_t = s\big]\Big]$$

$$= \mathbb{E}_{(\pi^*, \mathcal{M}) \sim \mathcal{P}(D,A)}\Big[1 - \widehat{\pi}_t(\pi_t^*(s_t)|s_t)\Big] \, \mathbb{E}_{(\pi^*, \mathcal{M}) \sim \mathcal{P}(D,A)}\Big[\Pr_{\widehat{\pi}}[\tau = t, s_t = s]\Big] \tag{126}$$

$$\stackrel{(a)}{=} \left(1 - \frac{1}{|\mathcal{A}|}\right) \mathbb{E}_{(\pi^*, \mathcal{M}) \sim \mathcal{P}(D,A)}\Big[\Pr_{\widehat{\pi}}[\tau = t, s_t = s]\Big] \tag{127}$$

where in $(a)$, conditioned on $(D, A)$ we use the fact that either $(i)$ $s = b$, in which case $\tau \neq t$ and both sides are 0, or $(ii)$ if $s \neq b$, then $\tau = t$ implies that the state $s$ visited at time $t$ must not be observed in $D$, so $\pi_t^*(s) \sim \text{Unif}(\mathcal{A})$. Using the fact that $\Pr_{\widehat{\pi}}[\tau_b = t + 1, \tau = t, s_t = s] \leq \Pr_{\widehat{\pi}}[\tau_b = t + 1, s_t = s]$ and summing over $s \in \mathcal{S}$ results in the inequality,

$$\mathbb{E}_{(\pi^*, \mathcal{M}) \sim \mathcal{P}(D,A)}\Big[\Pr_{\widehat{\pi}}[\tau_b = t + 1]\Big] \geq \left(1 - \frac{1}{|\mathcal{A}|}\right) \mathbb{E}_{(\pi^*, \mathcal{M}) \sim \mathcal{P}(D,A)}\Big[\Pr_{\widehat{\pi}}[\tau = t]\Big] \tag{128}$$

Multiplying both sides by $H - t$ and summing over $t = 1, \cdots, H$,

$$\mathbb{E}_{(\pi^*, \mathcal{M}) \sim \mathcal{P}(D,A)}\Big[\mathbb{E}_{\widehat{\pi}}[H - \tau_b + 1]\Big] \geq \left(1 - \frac{1}{|\mathcal{A}|}\right) \mathbb{E}_{(\pi^*, \mathcal{M}) \sim \mathcal{P}(D,A)}\Big[\mathbb{E}_{\widehat{\pi}}[H - \tau]\Big] \tag{129}$$

here we use the fact that the initial distribution $\rho$ places no mass on the bad state $b$. Therefore, $\Pr_{\widehat{\pi}(D)}[\tau_b = 1] = \rho(b) = 0$. This equation in conjunction with eq. (123) completes the proof.

### B.5.3 Proof of Lemma A.17

Firstly, in Lemma B.7 we show that $\mathbb{E}\big[\Pr_{\widehat{\pi}(D,A)}[\tau \leq \lfloor H/2 \rfloor]\big] \geq 1 - (1 - \gamma)^{\lfloor H/2 \rfloor}$ where $\gamma$ is defined as $\sum_{s \in \mathcal{S}} \rho(s)(1 - \rho(s))^N$. Subsequently, in Lemma B.8 we show that $\gamma \gtrsim |\mathcal{S}|/N$. Putting these two results together proves the statement of Lemma A.17.

Along the way to proving Lemma B.7, we introduce an auxiliary result.

**Lemma B.6.** *Fix the dataset $(D, A)$ collected by the learner, and any policy $\pi^* \in \Pi_{\text{mimic}}(D, A)$ (defined in eq. (36)). Recall that $\tau$ as defined in Lemma A.16 is the first time $t$ that the learner encounters a state $s_t \neq b$ that has not been visited in $D$ at time $t$.*

*For some $t \in [H]$, consider $\Pr_{\widehat{\pi}(D)}[\tau = t]$ computed with the underlying MDP as $\mathcal{M}[\pi^*]$. Then,*

$$\Pr_{\widehat{\pi}(D,A)}[\tau = t] = (1 - \rho(\mathcal{S}_t(D) \setminus \{b\})) \prod_{t'=1}^{t-1} \rho\big(\mathcal{S}_{t'}(D) \setminus \{b\}\big) \tag{130}$$

*Proof.* First observe that, the event $\{\tau = t\}$ implies that the learner only visits states in $\mathcal{S}_{t'}(D) \cup \{b\}$ till time $t' < t$, and visits a state in $\mathcal{S}_\tau(D) \cup \{b\}$ at time $t$. That is,

$$\Pr_{\widehat{\pi}}[\tau = t] = \Pr_{\widehat{\pi}}\Big[s_t \notin \mathcal{S}_t(D) \cup \{b\}, \; \forall t' < t, s_{t'} \in \mathcal{S}_{t'}(D) \cup \{b\}\Big] \tag{131}$$

$$= \Pr_{\widehat{\pi}}\Big[s_t \notin \mathcal{S}_t(D) \cup \{b\}, \; \forall t' < t, s_{t'} \in \mathcal{S}_{t'}(D) \setminus \{b\}\Big] \tag{132}$$

where in the last equation, we use the fact that by construction of $\mathcal{M}[\pi^*]$, the learner is forced to visit the state $b$ at time $t$ if the state $b$ is visited at any time $t' < t$.

Moreover, since the learner never visits $b$ till time $t - 1$, this implies that the learner must play the expert's action at each visited state until time $t - 1$ (otherwise the state $b$ is visited with probability 1 at time $t$). Therefore,

$$\Pr_{\widehat{\pi}}[\tau = t] = \Pr_{\pi^*}\Big[s_t \notin \mathcal{S}_t(D) \cup \{b\}, \; \forall t' < t, s_{t'} \in \mathcal{S}_{t'}(D) \setminus \{b\}\Big]. \tag{133}$$

Since under the policy $\pi^*$ rolled out on $\mathcal{M}[\pi^*]$, the distribution over states induced is i.i.d. across time and drawn from $\rho$, we have that,

$$\Pr_{\widehat{\pi}}[\tau = t] = (1 - \rho(\mathcal{S}_t(D) \cup \{b\})) \prod_{t'=1}^{t-1} \rho(\mathcal{S}_{t'}(D) \setminus \{b\}) \tag{134}$$

However the distribution $\rho$ has no mass on the state $b$. Therefore $\rho(\mathcal{S}_t(D) \cup \{b\}) = \rho(\mathcal{S}_t(D) \setminus \{b\})$ and the proof concludes. $\qquad\square$

**Corollary B.1.** $\Pr_{\widehat{\pi}(D,A)}[\tau \le \lfloor H/2 \rfloor] = 1 - \prod_{t=1}^{\lfloor H/2 \rfloor} \rho\big(\mathcal{S}_t(D) \setminus \{b\}\big).$

**Lemma B.7.** *Fix some policy* $\pi^* \in \Pi_{\mathrm{mimic}}(D, A)$ *and the MDP as* $\mathcal{M}[\pi^*]$. *Then,*

$$\mathbb{E}\big[\Pr_{\widehat{\pi}(D,A)}[\tau \le \lfloor H/2 \rfloor]\big] \ge 1 - (1-\gamma)^{\lfloor H/2 \rfloor} \tag{135}$$

*where* $\gamma = \sum_{s \in \mathcal{S}} \rho(s)(1 - \rho(S))^N$.

*Proof.* Recall that the learner rolls out policies $\pi_1, \cdots, \pi_N$ to generate trajectories $\mathsf{tr}_1, \cdots, \mathsf{tr}_N$. First observe that, conditioned on the learner's dataset truncated till the states visited at time $t$,

$$\mathbb{E}\Big[\prod_{t=1}^{\tau} \rho\big(\mathcal{S}_t(D) \setminus \{b\}\big)\Big] - \mathbb{E}\Big[\prod_{t=1}^{\tau+1} \rho\big(\mathcal{S}_t(D) \setminus \{b\}\big)\Big]$$
$$= \mathbb{E}\Big[\prod_{t=1}^{\tau} \rho\big(\mathcal{S}_t(D) \setminus \{b\}\big)\Big(1 - \mathbb{E}\big[\rho\left(\mathcal{S}_{\tau+1}(D) \setminus \{b\}\right) \big| D_{\le \tau, < \tau}\big]\Big)\Big] \tag{136}$$

where in the last equation we use the fact $\mathcal{S}_t(D)$ for all $t \le \tau$ is a measurable function of $D_{\le \tau, < \tau}$. Conditioned on $D_{\le \tau, < \tau}$, consider the distribution over actions $a_\tau^n$ played by the learner in different trajectories. If $a_\tau^n = \pi_t^*(s_\tau^n)$, the state $s_{\tau+1}^n$ is renewed in the distribution $\rho$. If $a_\tau^n$ is any other action, $s_{\tau+1}^n = b$ with probability 1, and does not provide any contribution to $\rho\left(\mathcal{S}_{\tau+1}(D) \setminus \{b\}\right)$. Let the random variable $N'$ denote the number of trajectories that have already visited $b$ prior to time $\tau$ or play an action other than the expert's action at time $\tau$. By linearity of expectation,

$$1 - \mathbb{E}\big[\rho\left(\mathcal{S}_{\tau+1}(D) \setminus \{b\}\right) \big| D_{\le \tau, < \tau}\big] = \mathbb{E}\Big[\sum_{s \in \mathcal{S} \setminus \{b\}} \rho(s)\left(1 - \rho(s)\right)^{N'} \Big| D_{\le \tau, < \tau}\Big] \tag{137}$$

$$\ge \sum_{s \in \mathcal{S} \setminus \{b\}} \rho(s)\left(1 - \rho(s)\right)^N \tag{138}$$

Recalling that $\gamma$ is defined as the constant $\sum_{s \in \mathcal{S}} \rho(s)\left(1 - \rho(s)\right)^N$ and $\rho(b) = 0$, from eqs. (136) and (138),

$$\mathbb{E}\Big[\prod_{t=1}^{\tau+1} \rho\big(\mathcal{S}_t(D) \setminus \{b\}\big)\Big] \ge (1 - \gamma)\mathbb{E}\Big[\prod_{t=1}^{\tau} \rho\big(\mathcal{S}_t(D) \setminus \{b\}\big)\Big] \tag{139}$$

We also have that $\mathbb{E}[\rho(\mathcal{S}_1(D) \setminus \{b\})] = 1 - \sum_{s \in \mathcal{S} \setminus \{b\}} \rho(s)(1 - \rho(s))^N = 1 - \gamma$ since the initial state $s$ in each trajectory in $D$ is sampled independently and identically from $\rho$. Using this fact and recursing eq. (139) over $\tau = 1, \cdots, \lfloor H/2 \rfloor - 1$ gives,

$$\mathbb{E}\Big[\prod_{t=1}^{\lfloor H/2 \rfloor} \rho\big(\mathcal{S}_t(D) \setminus \{b\}\big)\Big] \ge (1 - \gamma)^{\lfloor H/2 \rfloor}. \tag{140}$$

Invoking Corollary B.1 completes the proof. $\qquad\square$

**Lemma B.8.** $\gamma$, *defined in Lemma B.7 as* $\sum_{s \in \mathcal{S}} \rho(s)(1 - \rho(s))^N$ *is* $\ge \frac{|\mathcal{S}| - 2}{e(N+1)}$.

*Proof.* By the definition of $\rho$, we have that,

$$\gamma = \sum_{s \in \mathcal{S}} \rho(s)(1 - \rho(s))^N \overset{(i)}{\ge} \frac{|\mathcal{S}| - 2}{N + 1}\left(1 - \frac{1}{N + 1}\right)^N \ge \frac{|\mathcal{S}| - 2}{e(N + 1)}. \tag{141}$$

where in $(i)$ we lower bound by only considering the $|\mathcal{S}| - 2$ states having mass $= \frac{1}{N+1}$ under $\rho$. $\quad\square$

## B.6 Missing proofs for Theorem 5.1 (b)

### B.6.1 Proof of Lemma A.19

The proof of this result closely follows that of Lemma A.15. Fix some policy $\pi \in \Pi_{\mathrm{det}}$. Consider any time $t \in [H]$ and state $s \in \mathcal{S}_t(D)$ which is visited in some trajectory in the dataset at time $t$. If $\pi_t(s)$ does not match the unique action $a_t^*(s)$ played at time $t$ in any trajectory in $D$ that visits $s$ at this time, the likelihood of $\pi$ given $D$ is exactly 0 (recall we assume that the expert's policy is deterministic). On the contrary, the conditional probability of observing the expert dataset $D$ does not depend on the expert's action on the states that were not observed in $D$, since no trajectory visits these states. On these states the expert's action marginally follows the uniform distribution over $\mathcal{A}$. Thus the result follows.

### B.6.2    Proof of Lemma A.20

Observe that,

$$\mathbb{E}_{(\pi^*,\mathbf{r})\sim\mathcal{P}'(D)}\left[H - J_{\mathbf{r}}(\widehat{\pi}(D,P))\right]$$

$$= \mathbb{E}_{(\pi^*,\mathbf{r})\sim\mathcal{P}'(D)}\left[\mathbb{E}_{\widehat{\pi}}\left[\sum_{t=1}^{H} 1 - \mathbf{r}_t(s_t,a_t)\right]\right] \tag{142}$$

$$\geq \sum_{t=1}^{H} \mathbb{E}_{(\pi^*,\mathbf{r})\sim\mathcal{P}'(D)}\left[\mathbb{E}_{\widehat{\pi}}\left[\mathbb{1}(s_1 \notin \mathcal{S}_1(D))\left(1 - \mathbf{r}_t(s_t,a_t)\right)\right]\right] \tag{143}$$

By construction of the $\mathcal{M}[\pi^*]$ and $P$ each state $s \in \mathcal{S}$ is absorbing. Therefore, $s_1 \notin \mathcal{S}_1(D) \iff \{\forall t \in [H], s_t \notin \mathcal{S}_t(D)\}$. By the structure of the reward function $\mathbf{r}[\pi^*]$, the learner accrues a reward of 1 at some state if and only if the learner plays the expert's action at this state. Therefore, $\mathbf{r}_t(s_t,a_t) = \mathbb{1}(a_t = \pi_t^*(s_t))$ and,

$$\mathbb{E}_{(\pi^*,\mathbf{r})\sim\mathcal{P}'(D)}\left[\mathbb{E}_{\widehat{\pi}}\left[\mathbf{r}_t(s_t,a_t)\Big|s_1 \notin \mathcal{S}_1(D)\right]\right]$$

$$= \mathbb{E}_{(\pi^*,\mathbf{r})\sim\mathcal{P}'(D)}\left[\mathbb{E}_{\widehat{\pi}}\left[\mathbb{1}(a_t = \pi_t^*(s_t))\Big|s_1 \notin \mathcal{S}_1(D)\right]\right] \tag{144}$$

From Lemma A.19 observe that conditioned on $D$, the expert's policy $\pi^*$ is sampled uniformly from $\Pi_{\mathrm{mimic}}(D)$. Since we condition on $s_1 \notin \mathcal{S}_1(D) \iff s_t \notin \mathcal{S}_t(D)$ the state $s_t$ is not visited in any trajectory in $D$ at time $t$. This implies that the expert's action $\pi_t^*(s_t)$ is uniformly sampled from $\mathcal{A}$. Therefore,

$$\mathbb{E}_{\widehat{\pi}}\left[\mathbb{E}_{(\pi^*,\mathbf{r})\sim\mathcal{P}'(D)}\left[\mathbb{1}(a_t=\pi_t^*(s_t))\Big|s_1 \notin \mathcal{S}_1(D)\right]\right] = \frac{1}{|\mathcal{A}|}\sum_{a\in\mathcal{A}}\mathbb{E}_{\widehat{\pi}}\left[\mathbb{1}(a_t=a)\Big|s_1 \notin \mathcal{S}_1(D)\right] = \frac{1}{|\mathcal{A}|}.$$

Plugging this into eq. (144) and subtracting 1 from both sides we get that,

$$\mathbb{E}_{(\pi^*,\mathbf{r})\sim\mathcal{P}'(D)}\left[\mathbb{E}_{\widehat{\pi}(D,P)}\left[1 - \mathbf{r}_t(s_t,a_t)\Big|s_1 \notin \mathcal{S}_1(D)\right]\right] = 1 - \frac{1}{|\mathcal{A}|}. \tag{145}$$

Plugging this back into eq. (143) we get that,

$$\mathbb{E}_{(\pi^*,\mathbf{r})\sim\mathcal{P}'(D)}\left[H - J_{\mathbf{r}}(\widehat{\pi}(D,P))\right] \geq H\left(1 - \frac{1}{|\mathcal{A}|}\right)\mathrm{Pr}_{\widehat{\pi}(D,P)}\left[s_1 \notin \mathcal{S}_1(D)\right] \tag{146}$$

Since $s_1$ is sampled independently from $\rho$, the proof of the result concludes.

### B.6.3    Proof of Lemma A.21

Note that the dataset $D$ follows the posterior distribution generated by rolling out $\pi^*$ for $N$ episodes when $\pi^*$ is drawn from the uniform prior $\mathrm{Unif}(\Pi_{\mathrm{det}})$. Irrespective of the choice of $\pi^*$, note that the initial distribution over states is still $\rho$. Therefore,

$$\mathbb{E}[1 - \rho(\mathcal{S}_1(D)) = \sum_{s\in\mathcal{S}}\rho(s)(1-\rho(s))^N \tag{147}$$

$$\overset{(i)}{\geq} \frac{|\mathcal{S}|-1}{N+1}\left(1 - \frac{1}{N+1}\right)^N \geq \frac{|\mathcal{S}|-1}{e(N+1)} \tag{148}$$

where in $(i)$ we lower bound by considering only the $|\mathcal{S}| - 1$ states having mass $\frac{1}{N+1}$ under $\rho$. Plugging this back into eq. (57) completes the proof of the theorem.