[Reviews · NeurIPS 2020]

Review 1

Summary and Contributions: This paper presents a number of theoretical results for (variants of) behavioural cloning approaches to imitation learning. These results indicate that the strategy of directly mimicking the expert is condemned in the worst-case to make an error that scales quadratically with the horizon. The same applies to a variant involving querying the expert, e.g. Dagger. It seems that presented upper bounds on these methods are similar to those known in the literature, e.g. RGB11. Another result of the paper is that an improvement to linear dependency in horizon can be obtained by making a transition function available to the learner. I believe this result is meaningful as a number of practical approaches are based on the use of simulator to overcome the expert-learner distribution shift. ----------- Rebuttal ----------- I have read the author feedback. * I believe that the problem characterization in terms of number of states |S| is weakly informative. In high-dimensional spaces, it should be expected to find an error in close proximity of the expert trajectory. I would be interested in worst-case results over distribution shifts that are likely to occur in practice; e.g., in the robust MDP framework one solves a minimax problem over a *bounded* amount of noise (see e.g. G. Iyengar "Robust dynamic programming"). * In the light of previous comment, it is not clear to me that this is a limitation of the imitation learning. * Given that quadratic upper bounds for supervised learning algorithm are tight for deterministic expert policies, the proposed lower bound (Theorem 4) is not surprising. * The real issue is how to recover from the unseen situations. This paper proposes the use of transition / simulator that seems to me much more informative result, unfortunately, only limited to one section. Due to the above considerations, I do not revise my score.

Strengths: I find the analysis with the known transition matrix informative on how to tackle the distribution mismatch between the learner and the expert in behavioral cloning. It seems that adjusting for covariate shift using a simulator / transition function is a promising approach. Practical applications would be benefit from potential extension of these results to a setting with approximate transition function.

Weaknesses: The simple behaviour cloning as well as interactive approaches are analysed in the literature using reduction to supervised learning. Thus, somewhat unsurprisingly, they suffer from the same caveats in the presence of distribution shift. When the state space is large, it will always be possible to find a state that wasn't visited by the expert trajectory. Thus, I am not fully convinced of the implications of the worst-case results presented in the paper. In addition, some results only apply to deterministic expert policy that I find restrictive in a practical setting.

Correctness: It seems that the upper bounds on behavioral cloning (simple and interactive) are in-line with the previous work. The lower bound results intuitively correspond to the supervised learning reduction. For the known-transitions setting, the theorem statements make sense, although I did not check the details of the proofs.

Clarity: The paper is well-structured by first presenting the overview of results and then describing the details. It would be helpful for the reader to visually illustrate the Mimic-MD algorithm as its mathematical notation is quite dense.

Relation to Prior Work: It would be helpful to include in Table 1 the comparison of the obtained bounds to the prior work. I feel that this work would benefit from deeper relating to the previously proposed practical algorithms and their theoretical analysis, e.g. SVBB19.

Reproducibility: Yes

Additional Feedback:


Review 2

Summary and Contributions: In this paper, the authors study theoretical bounds for imitation learning algorithms, with a focus on behavior cloning, in various settings including no-interactions, interactive and known dynamics. A major result is a quadratic term in time horizon is a fundamental gap in imitation learning, whatever the underlying algorithm is. In settings where dynamics are known, the quadratic barrier can be broken via the proposed MIMC-MD algorithm. Finally, lower bounds are given are these settings as well. ====== Post-rebuttal: thank you for the rebuttal. I still hope there were more discussions on DAgger vs BC. One potential improvement is to provide concrete assumptions on MDP where one could separate DAgger and BC on sample complexity. As a result, I decided to keep my original score.

Strengths: The presentation of theoretical results is very well done, especially the transition between each setting and the connections between them. The results in Section 4 on behavior cloning, while not groundbreaking, help unify known results. Together with the lower bounds in Section 6, they establish the optimality of behavior cloning in terms of the minimax expected error, which is a somewhat surprising result. The result in Section 5, where the dynamics are assumed known, is significant in achieving the first bound to break the quadratic compounding errors. These results are very relevant to the NeurIPS community as they provide theoretical insights to general imitation learning algorithms.

Weaknesses: I found the study on the active setting lacking. In Table 1, it is stated an upper bound is provided for active setting in Theorem 1 but the actual Theorem 1 only considers the no-interaction setting. While it is not entirely surprising that the active setting does not improve upon behavior cloning in the worst case, some discussion on their empirical success, as demonstrated by algorithms such as DAgger, will help bridge the gap between no-interaction and active settings.

Correctness: The claims seem correct though I did not check the proofs in the Appendix.

Clarity: This paper is mostly clear. In equation (9), the probability terms are not defined, though one can mostly understand what they mean with the context.

Relation to Prior Work: The differences from previous works are stated clearly.

Reproducibility: Yes

Additional Feedback: In Section 5, Remark 2, the authors mention various distribution matching approaches in imitation learning. Algorithm 2 can be also viewed as a version of distribution matching (equation (8)). Can the authors provide some intuition on what distributions are being matched? And what is the basis for the conjecture that existing distribution matching approaches are unlikely to achieve the bound in Theorem 3?


Review 3

Summary and Contributions: The paper studies the statistical sub-optimality bounds (w.r.t expert's value) of imitation learning algorithms in three settings of episodic tabular MDPs: (a) no-interaction with MDP allowed but a dataset of N expert trajectories is provided, (b) known-transition: MDP transition function is provided in additional to (a), and (c) active: no dataset provided but MDP interactions allowed and expert queries allowed at any state. Upper and lower bounds are presented (mostly for behavior cloning) for the two cases of deterministic and non-deterministic experts, out of which the the upper bound on sub-optimality for non-deterministic experts by an empirical risk minimizer under log-loss is a novel contribution as it does not depend on the cardinality of the action space (Theorem 3 in paper). Further, in the known-transition setting, a lower bound is provided which depends linearly on episode length H (i.e. |S| H / N) and a novel algorithm MIMIC-MD is proposed that achieves a (approx.) |S| H^{3/2} / N upper bound on sub-optimality, improving upon that of max likelihood estimation (MLE) and the traditional |H|^2 dependence of sub-optimality due to compounding errors in the other unknown-transition settings. The intuition of the algorithm is that even if a mistake is made by the learner that leads it to states not present in the expert's demonstrations, it can correct itself since it knows the transition function. Such corrections are approximately computed by dividing the expert dataset into half, the first half being used for imitating the expert and the other for approximating the probability of correction. ==================================== Post-rebuttal update: After carefully going through the other reviews (especially the comments by R1 and R4) and the author's response, it is clear that I had missed several drawbacks of the paper in my original assessment (because of my unfamiliarity with relevant prior work), particularly the lack of significance of the lower bounds (in Theorem 4), computational inefficiency of the algorithm (while also acknowledging the contribution, which will help future work in reducing sample complexity) and the lack of discussion about recovery from unseen situations (R1). Out of these, I have considered (to the best of my knowledge) the first to be important while the second and third drawback less relevant to the main contribution (and assumptions) of the paper. However, I do not feel that these drawbacks are sufficient grounds for decreasing my rating, so I will maintain it and recommend acceptance. In order to reflect the updated confidence of my assessment, I have additionally reduced my confidence score from 3 to 2.

Strengths: The paper's strengths are in the novel algorithms or bounds proposed in the imitation learning settings: - In the no-interaction setting (dataset of expert demonstrations provided), behavior cloning is shown to be optimal for deterministic experts and empirical risk minimization under log-loss is optimal for stochastic experts. The latter is the suggested to be the first such bound without dependence on the cardinality of the action space. - Due to the optimality of behavior cloning and the lower bound in Theorem 1, it is shown that methods in the active setting (can query expert and interact with MDP) cannot improve over behavior cloning in the no-interaction setting (for e.g.: for active setting algorithms such as DAGGER, Aggravate). - The MIMIC-MD algorithm is proposed that uses the intuition that corrections can be made in the known-transition setting to get back to states in the expert's demonstrations after making an error. This algorithm is shown to have a sub-optimality upper bound with |H| raised to the power 3/2, which is shown to be just a square root of |H| times away from the worst case lower bound of |H|.

Weaknesses: Nothing notable to the best of my knowledge.

Correctness: The proof sketches throughout the paper are quite accessible and seem reasonably correct. However, I could not verify the correctness of all the proofs in the appendix.

Clarity: The paper is well written, with a good slow build up of informal theorems highlighting the main contributions to the final core results.

Relation to Prior Work: Yes, the paper is appropriately discussed with respect to prior work.

Reproducibility: Yes

Additional Feedback:


Review 4

Summary and Contributions: This work provides a thorough and clear summary of lower and upper bounds in three related problems in imitation learning over finite MDPs with bounded rewards: no-interaction (which is classical), active (which is similar to that in DAgger [RGB11]), and known transition (a novel setting). Though some results are similar to prior works ([RB10]), or perhaps not very surprising for some, the thoroughness is remarkable. The specific novel contributions include: 1. Making clear that without further assumptions (like that in DAgger), the lower bound of active setting matches the upper bound of behavior cloning in the no-interaction setting, which are both O(H^2). 2. Pointing out that the reduction to supervised learning approach has limitations when the expert policy is stochastic (in the form of a dependency on A and a slow statistical convergence of √N). 3. Proposing the known transition setting and an algorithm MIMIC-MD (not based on reduction) which has an expected error of O(H^(3/2)).

Strengths: 1. Answer many basic questions regarding the relations of two popular settings in IL (See Summary1). 2. Propose a new setting (known transition) that captures intuition about the learner’s ability of correcting mistakes. Moreover, a near-optimal algorithm is proposed. 3. The writing of the main article body is easy to follow with substantive intuitions.

Weaknesses: 1. A discussion of the computational complexity of MIMIC-MD would be a nice addition. What is it? Is there a nice procedure for the optimization problem (8)?

Correctness: The article is theoretical and the results seem correct upon high-level checks.

Clarity: The presentation is excellent. I enjoyed reading the article and its supplement.

Relation to Prior Work: The rigor and clarity of this work contrasts favorably to other works in this domain in my opinion. 1. [RB10] contains in its supplement a lower bound example for the no-interaction setting which yields O(H^2). How would you compare your lower bound example (Fig 1(a)) with theirs? In particular, does the example of [RB10] undermines your claim on L84?

Reproducibility: Yes

Additional Feedback: 1. Could you comment on the obtained lower/upper bounds of the known transition setting? Do you think either the example is weak or the analysis of MIMIC-MD is loose? It may help to provide some numerical results. 2. (Minor) (54) in Appendix B.1.1 should be an equality, right? And the constant in (55) can be made 1/e (which is smaller than 4/9 and nicer), c.f., (135). Several results seem to depend essentially on Lemma 1. Perhaps it is worth mentioning in the main text? 3. (Minor) L565 typo. T^\pi_{t, s}(i) —> T^*_{t, s}(i). 4. Inhomogeneous (or episodic) MDPs are common in IL literature but their results often lack finer instance-dependency on the Markov structure unlike RL results on homogenous MDPs (where transitions are time-invariant). Given that your lower bound examples are homogeneous MDPs, could you comment on whether some results would be different under homogeneous MDPs (if space permits). ## Post-rebuttal Thank you for replying and commenting. I slightly lowered my score (9 -> 8) to reflect the new information on the computational inefficiency of MIMIC-MD which also explains the absence of numerical confirmation. I will suggest making this fact explicit in any future version as "algorithm" without further qualifications is usually understood as computationally efficient. As R1 has touched on, a discussion/illustration on recovery (under a good mimic policy according to (8)) may provide readers insights to a computationally efficient approximation (for future studies).

[Author Response · NeurIPS 2020]

We thank the reviewers for the detailed feedback. Before addressing the comments, we acknowledge the typographic /
clarification errors, the scope for improvement in the constants, and in the subsequent version of the paper will include
a comparison with existing work (such as [SVBB19]) in Table 1. Below we address each reviewer's comments.

**R1:** We analyze the standard behavior cloning (BC) approach to give a worst-case rate at which its error goes to 0
as a function of $N$ (size of expert dataset), as $\lesssim |\mathcal{S}|H^2/N$. Although this rate largely follows from existing work, we
only state this as an achievability counterpart for our more important contribution here: to establish a universal lower
bound of $\gtrsim |\mathcal{S}|H^2/N$. We show that for *any algorithm* (even if it can actively query the expert) there exists an instance
on which large error ($\gtrsim |\mathcal{S}|H^2/N$) must be incurred. This compounding error lower bound does not have anything in
particular to do with BC, and uniformly applies for any learner algorithm. In contrast, the lower bound example of
[RB10] (**mentioned by R4**) applies only for supervised learning. They construct a particular MDP and show that a
particular learner strategy which plays an action different than the expert with probability $\epsilon$ has error $\gtrsim H^2\epsilon$. It turns
out, in their instance, the error incurred by BC is *exactly* 0 *given just a single expert trajectory*. Thus it does not imply a
uniform lower bound on the error of all learner algorithms as a function of $N$: even BC performs well on their example.

*Comparison with* FAIL *[SVBB19]:* In the det. expert setting, without additional assumptions, the worst case error
guarantee of BC is superior to FAIL. In [Theorem 3.3, SVBB19] choosing $\Pi$ to be the set of all deterministic policies
(of size $|\mathcal{A}|^{|\mathcal{S}|}$) shows that FAIL achieves error $\sqrt{|\mathcal{S}||\mathcal{A}|H^5/N}$ (ignoring log-factors). In contrast, we show that behavior
cloning incurs error $|\mathcal{S}|H^2/N$ [1]. This is always better: not only is it independent of $|\mathcal{A}|$, but has optimal dependence on
$H$ and $N$. However, we clarify that FAIL also applies in the ILFO setting where the expert actions are not observed.
Furthermore, if the expert is non-deterministic, we show that MIMIC-EMP has expected error $|\mathcal{S}|H^2/N$ (ignoring
log-factors). This again significantly improves on FAIL and surprisingly is independent of $|\mathcal{A}|$. We emphasize that the
proof of this result is quite involved and uses a particular coupling based argument - as discussed in the paper, applying
simple reduction based analyses is loose, failing to avoid dependence on $|\mathcal{A}|$ and converging slowly at a $1/\sqrt{N}$ rate.

Although some of our results apply only in the det. expert setting, we appeal that in single agent RL every optimal
policy is deterministic. Thus, studying the case where the expert policy is deterministic is not too restrictive as it
includes the best possible expert policy. Another critique is that formulating IL as a minimax problem and studying
worst-case guarantees might be too pessimistic. We argue that studying the minimax approach is a basic formulation for
such statistical problems, and provides a benchmark for further improvements.

**R2:** in this context, by studying in the minimax framework, a conclusion of our work is that additional assumptions
on the MDP / reward structure are necessary for an active query algorithm such as DAgger to outperform BC, and
explain its superior empirical performance. We believe that the assumptions imposed are critical: weak assumptions
may not be able to separate the sample complexity in the active and no-interaction settings, while strong assumptions
may compromise the practical relevance of obtained results. Also, to clarify, the BC error upper bound in Theorem 1
translates to the active setting by a modification to the algorithm: by playing the expert's action at visited states when
interacting with the MDP, the learner can generate $N$ expert trajectories; then the learner performs BC on this dataset.

Next, we give an intuition for MIMIC-MD in the known transition setting: MIMIC-MD copies the expert action on
states seen in the dataset, so the learner incurs error only upon visiting new states (i.e. not seen in the expert dataset). So
the appropriate quantity to match is the probabilities induced over states and actions by the expert, when at some point
in the episode a new state is visited. Data splitting can in fact be used to estimate these probabilities which immediately
leads to the form of MIMIC-MD. That said, it is unclear how to efficiently carry out simulations to identify whether
MIMIC-MD admits better error guarantees (**as suggested by R4**). The error incurred by MIMIC-MD (or any policy
for that matter) requires evaluating against its corresponding worst MDP instance, one among exponentially many
instances. We believe this is an important open problem to resolve in the context of the known-transition setting.

The basis for conjecturing that existing distribution matching approaches are unlikely to be optimal is because they do
not take into account that the expert's actions are known at all states seen in the dataset. These policies may choose to
play a different action at a state, even if the expert's action is observed in the dataset. In contrast, MIMIC-MD returns a
policy that is constrained to mimic the expert at states visited in the expert dataset, and avoids this problem.

**R4:** MIMIC-MD is not a polynomial-time algorithm as the optimization in Eq. (8) is over multivariate degree-$H$
polynomials. An important future work is to translate this intuition to a polynomial-time algorithm - perhaps one which
returns a policy which approximately solves Eq. (8). We also add that the lower bound instances in the known-transition
setting showing that the error of any policy is $\gtrsim |\mathcal{S}|H/N$ are indeed homogeneous MDPs. In fact, the time-variance
of the reward function can also be lifted if the action space is large enough (say $\gtrsim H$). On the other hand, in the
no-interaction setting our lower bounds are inhomogeneous. Here, it is an interesting question to design an algorithm
which leverages time-invariant transitions to improve on the $\lesssim |\mathcal{S}|H^2/N$ error rate of BC.

## Footnotes

[1] We plan to include an improvement to Theorem 1: we improve the $1/\sqrt{N}$ dependence in the high probability term in Eq. (4) to $\sqrt{|\mathcal{S}|}/N$. Thus BC enjoys a $1/N$ worst-case error rate, both in expectation as well as with high probability.


[Meta-Review · NeurIPS 2020]

The paper considers the theoretical study of imitation learning algorithms, with a focus on behavior cloning (BC) and the non-interacting setting (where the data is assumed fixed and no further data-collection is possible). The theoretical analysis focuses on minimax bounds, leading to a best-case and worst case analysis, the results are interesting (even if some of the theorems seem natural to hold) and make a contribution towards the understanding of the purely 'offline'/non-interaction setting; which is an active topic of research in the NeurIPS community. The reviewers all agreed that the setting is important, that the paper is well written, and that the theoretical results are new and interesting (even if there was some discussion about the strength of the bounds). There was a discussion among the reviewers regarding the derived bounds (and how informative they are in practice) that did not end in complete agreement among the reviewers (I summarize below). This leaves the paper in a somewhat borderline position. I am however of the opinion that, while the criticisms raised by reviewer #1 are valid, the paper makes enough of a contribution to be interesting to a large part of the NeurIPS community. As a result I recommend acceptance. Reasoning for decision and some points that should be clarified in the final version: 1) The provided presented bounds are worst and best-case analysis and are in terms of the state-space |S|, I agree with Reviewer #1 that this is limiting and I think this should be mentioned a bit more prominently in the paper, e.g. in the section where it is established that BC is optimal in terms of worst-case performance, and even dagger cannot improve upon this worst case bound, it would be good to explicitly note that this does not entail that no improvement is possible in expectation. Despite this, I think the results in Section 4, while intuitive, are interesting to the community as they neatly unify existing results and together with the lower bounds from Section 6 give optimality results for BC that I and the reviewers had not seen proven before. Irregardless of whether we believe these bounds to be very strong I do believe they will encourage new research and serve as a good reference for the community. 2) The section on the active-setting is somewhat less thorough in the paper. It feels as if part could be slightly de-emphasized to give the rest of the paper more room to breathe (especially the part on using a transition model, which has intersting results but is fairly short) 3) It should be clarified that MIMIC-MD is not a computationally efficient algorithm, and that developing such an efficient procedure from the general strategy outlined by it is an important step for future work. I do not think the fact that the algorithm is not efficient hampers the paper in any way, but it should be stated clearly.